# BUB-1 targets PP2A:B56 to regulate chromosome congression during meiosis I in *C. elegans* oocytes

Laura Bel Borja[1], Flavie Soubigou[1], Samuel J P Taylor[1], Conchita Fraguas Bringas[1], Jacqueline Budrewicz[2,3], Pablo Lara-Gonzalez[2,3], Christopher G Sorensen Turpin[4], Joshua N Bembenek[5], Dhanya K Cheerambathur[6], Federico Pelisch[1]*

[1]Centre for Gene Regulation and Expression, Sir James Black Centre, School of Life Sciences, University of Dundee, Dundee, United Kingdom; [2]Ludwig Institute for Cancer Research, San Diego, United States; [3]Department of Cellular and Molecular Medicine, University of California, San Diego, San Diego, United States; [4]Department of Biochemistry, Cellular and Molecular Biology, University of Tennessee, Knoxville, United States; [5]Department of Molecular, Cellular, and Developmental Biology, University of Michigan, Ann Arbor, United States; [6]Wellcome Centre for Cell Biology & Institute of Cell Biology, School of Biological Sciences, The University of Edinburgh, Edinburgh, United Kingdom

*For correspondence:
f.pelisch@dundee.ac.uk

**Competing interests:** The authors declare that no competing interests exist.

**Abstract** Protein Phosphatase 2A (PP2A) is a heterotrimer composed of scaffolding (A), catalytic (C), and regulatory (B) subunits. PP2A complexes with B56 subunits are targeted by Shugoshin and BUBR1 to protect centromeric cohesion and stabilise kinetochore–microtubule attachments in yeast and mouse meiosis. In *Caenorhabditis elegans*, the closest BUBR1 orthologue lacks the B56-interaction domain and Shugoshin is not required for meiotic segregation. Therefore, the role of PP2A in *C. elegans* female meiosis is unknown. We report that PP2A is essential for meiotic spindle assembly and chromosome dynamics during *C. elegans* female meiosis. BUB-1 is the main chromosome-targeting factor for B56 subunits during prometaphase I. BUB-1 recruits PP2A:B56 to the chromosomes via a newly identified LxxIxE motif in a phosphorylation-dependent manner, and this recruitment is important for proper chromosome congression. Our results highlight a novel mechanism for B56 recruitment, essential for recruiting a pool of PP2A involved in chromosome congression during meiosis I.

## Introduction

Formation of a diploid embryo requires that sperm and egg contribute exactly one copy of each chromosome. The cell division in charge of reducing ploidy of the genome is meiosis, which involves two chromosome segregation steps after a single round of DNA replication (*Marston and Amon, 2004*; *Ohkura, 2015*). Female meiosis is particularly error prone (*Hassold and Hunt, 2001*), which can lead to chromosomally abnormal embryos. Therefore, understanding the molecular events that guarantee proper chromosome segregation during female meiosis is of paramount importance. Cell division is under tight control of post-translational modifications (PTMs), of which phosphorylation is the most studied. The balance between kinase and phosphatase activities plays a central role, but we still lack a clear picture of how this is achieved during meiosis, especially when compared to mitosis (*Gelens et al., 2018*; *Novak et al., 2010*).

Protein Phosphatase 2A (PP2A) is a heterotrimeric serine/threonine phosphatase composed of a catalytic subunit C (PPP2C), a scaffolding subunit A (PPP2R1), and a regulatory subunit B (PPP2R2–

PPP2R5) (*Cho and Xu, 2007*; *Xu et al., 2008*; *Xu et al., 2006*). While the core enzyme (A and C subunits) is invariant, diversity in PP2A holoenzyme composition arises from the different regulatory B subunits. Four families of B subunits have been characterised: B55 (B), B56 (B′), PR72 (B″), and Striatin (B‴) (*Moura and Conde, 2019*; *Seshacharyulu et al., 2013*; *Shi, 2009*). In mammals and yeast, PP2A:B56 regulates the spindle assembly checkpoint (SAC) (*Espert et al., 2014*; *Hayward et al., 2019*; *Qian et al., 2017*; *Vallardi et al., 2019*), chromosome congression (*Xu et al., 2013*; *Xu et al., 2014*), and centromeric cohesion (*Kitajima et al., 2006*; *Riedel et al., 2006*; *Tang et al., 2006*).

PP2A:B56 can be targeted to distinct sites by different proteins, and two mechanisms have been characterised at the structural level. While an N-terminal coiled coil domain in Shugoshin/MEI-S332 binds PP2A:B56 (*Xu et al., 2009*), other substrates and/or regulatory proteins contain short linear motifs (SLiMs) following the consensus LxxIxE, which interact directly with B56 subunits (*Hertz et al., 2016*; *Van Roey and Davey, 2015*; *Wang et al., 2016*). BubR1 is one of the most widely studied LxxIxE-containing proteins and has been well characterised during mitosis, where it plays a role in targeting PP2A:B56 to kinetochores (*Foley et al., 2011*; *Kruse et al., 2013*; *Suijkerbuijk et al., 2012*). BubR1 is also required during mouse meiosis (*Homer et al., 2009*; *Touati et al., 2015*; *Yoshida et al., 2015*) where kinetochore–microtubule attachments are stabilised through kinetochore dephosphorylation by PP2A:B56 (*Yoshida et al., 2015*).

In *Caenorhabditis elegans*, the meiotic role of PP2A remains unexplored. The core components of the PP2A holoenzyme in *C. elegans* are LET-92 (catalytic C subunit) and PAA-1 (scaffolding A subunit), and the regulatory B subunits are B55[SUR-6] (*Sieburth et al., 1999*), B56[PPTR-1] and B56[PPTR-2] (*Padmanabhan et al., 2009*), B72[RSA-1] (*Schlaitz et al., 2007*), and CASH-1/Striatin (*Pal et al., 2017*). PP2A complexes containing B55[SUR-6] and B72[RSA-1] play important roles during mitosis (*Kitagawa et al., 2011*; *Schlaitz et al., 2007*; *Song et al., 2011*), and while PP2A likely plays a role during meiosis (*Schlaitz et al., 2007*), this has not been studied. Interestingly, the closest *C. elegans* BubR1 orthologue, Mad3[SAN-1], does not localise to unattached kinetochores (*Essex et al., 2009*) and lacks the domain responsible for PP2A:B56 targeting. Additionally, the *C. elegans* Shugoshin orthologue, SGO-1, is dispensable for protection of cohesion in meiosis I (*de Carvalho et al., 2008*) and for counteracting Aurora B-targeting histone H3T3 phosphorylation (*Ferrandiz et al., 2018*). These observations suggest the existence of additional unknown mechanisms of PP2A regulation.

In the present work, we used *C. elegans* oocytes to establish the role and regulation of PP2A during female meiosis. We found that PP2A is essential for female meiosis, and its recruitment to meiotic chromosomes and spindle is mediated by the B56 regulatory subunits PPTR-1 and PPTR-2. Targeting of the B56 subunits is mediated by the kinase BUB-1 through a canonical B56 LxxIxE motif, which targets both PPTR-1 and PPTR-2 to the chromosomes and central spindle during meiosis. Overall, we provide evidence for a novel, BUB-1-regulated role for PP2A:B56 in chromosome congression during female meiosis in *C. elegans*.

## Results

### PP2A is essential for spindle assembly and chromosome segregation during meiosis I

We used dissected *C. elegans* oocytes to assess the role and regulation of PP2A (see *Figure 1A* for schematic) during female meiosis by following spindle and chromosome dynamics using GFP-tagged tubulin and mCherry-tagged histone. During meiosis I, six pairs of homologous chromosomes (bivalents) are captured by an acentrosomal spindle. Chromosomes then align and segregate. For our analysis, we defined metaphase I (t = 0) as the frame before we detected chromosome separation (*Figure 1B*). In wild-type oocytes, chromosome segregation is associated with dramatic changes in microtubule organisation as anaphase progresses, with microtubule density decreasing in poles and the bulk of GFP::tubulin is detected in the central spindle (*Figure 1C*; *Figure 1—video 1*). In contrast, depletion of the PP2A catalytic subunit LET-92 drastically affects meiosis I with chromosomes failing to align in 84% of the oocytes (*Figure 1C*, cyan arrow; *Figure 1D*; p<0.0001, Fisher's exact test). During anaphase, chromosomes collapse into a small area where, in some cases, two chromosome masses were visible (*Figure 1C*, yellow arrow). Interestingly, severe chromosome segregation defects do not necessarily lead to polar body extrusion (PBE) defects (*Schlientz and Bowerman, 2020*). In the absence of PP2Ac, only 15% of oocytes achieved PBE (*Figure 1E*, p<0.0001, Fisher's

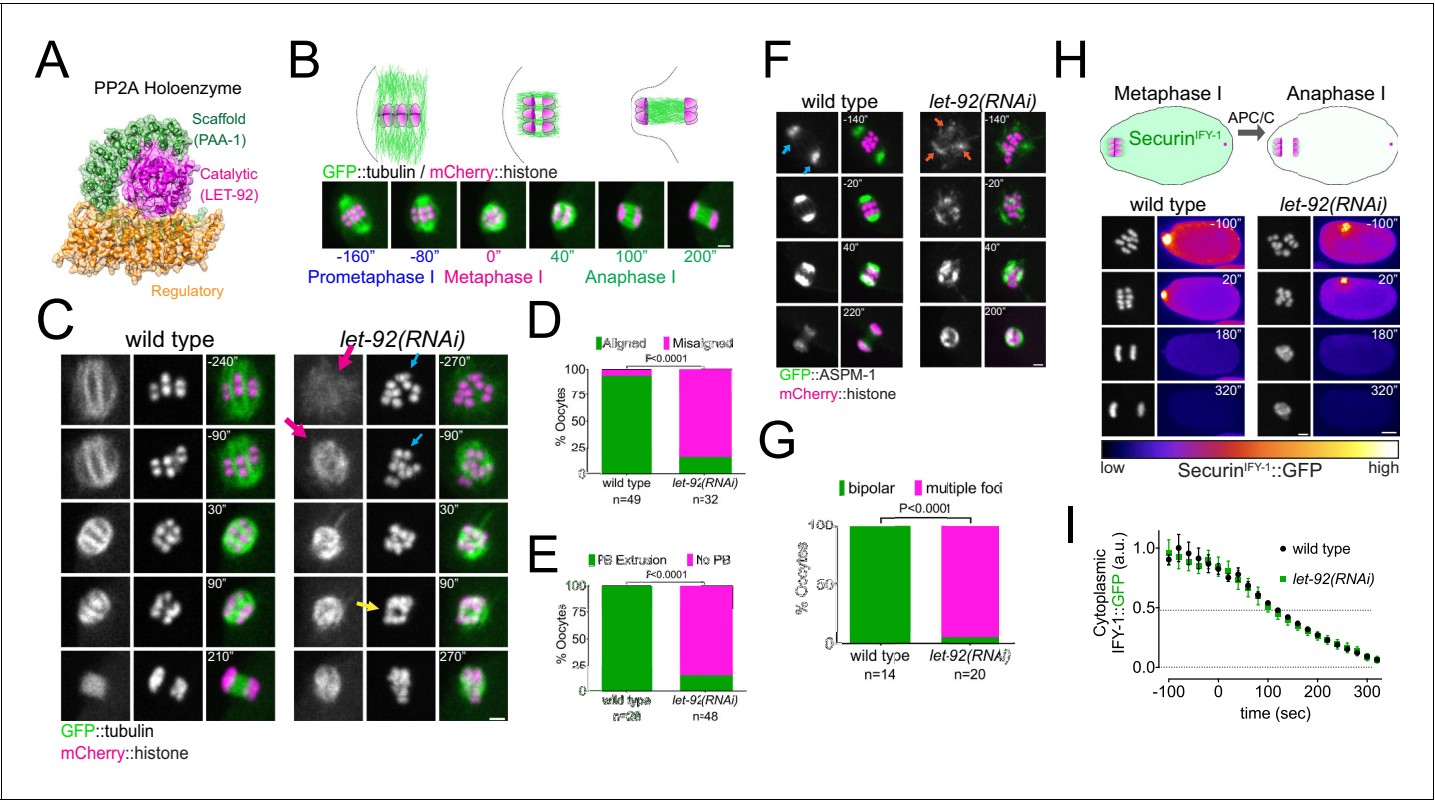

**Figure 1.** Protein Phosphatase 2A (PP2A) is essential for meiosis I in *C. elegans* oocytes. (**A**) Schematic of the PP2A heterotrimer highlighting the single catalytic and scaffold subunits in *C. elegans* (LET-92 and PAA-1, respectively). The schematic was generated from the PDB structure 2npp (*Xu et al., 2006*). (**B**) Schematic of the timescales used throughout the paper. Metaphase I was defined as time zero and chosen as the frame prior to the one where chromosome separation was detected. (**C**) Microtubule and chromosome dynamics were followed in wild-type and *let-92(RNAi)* oocytes expressing GFP::tubulin and mCherry::histone. Magenta arrows point to defective spindle structure; the cyan arrow points to misaligned chromosomes; the yellow arrow shows an apparent separation between chromosome masses. Inset numbers represent the time relative to metaphase I in seconds. Scale bar, 2 μm. See *Figure 1—video 1*. (**D**) The number of oocytes with misaligned chromosomes in metaphase I was compared between wild-type and *let-92(RNAi)* oocytes and the percentage is represented (p<0.0001, Fisher's exact test). The number of oocytes analysed (n) is shown. (**E**) The number of oocytes with an extruded polar body (PB) after meiosis I in wild-type and *let-92(RNAi)* oocytes was analysed and the percentage is represented (p<0.0001, Fisher's exact test). The number of oocytes analysed (n) is shown. (**F**) ASPM-1 and chromosome dynamics were followed in wild-type and *let-92 (RNAi)* oocytes expressing GFP::ASPM-1 and mCherry::histone. Inset numbers represent the time relative to metaphase I in seconds. Cyan arrows point to spindle poles, whereas orange arrows highlight the unfocused ASPM-1 cumuli. Scale bar, 2 μm. See *Figure 1—video 2*. (**G**) The number of oocytes with defective spindle at prometaphase I in wild-type and *let-92(RNAi)* oocytes was analysed, and the percentage is represented (p<0.0001, Fisher's exact test). The number of oocytes analysed (n) is shown. (**H**) Securin[IFY-1] degradation was used as a proxy for anaphase progression and followed in wild-type and *let-92(RNAi)* oocytes. Greyscale images of the chromosomes are shown as well as whole oocyte images using the 'fire' LUT. The intensity scale is shown in the bottom. Inset numbers represent the time relative to metaphase I in seconds. Scale bar, 2 μm. See *Figure 1—video 3*. (**I**) Cytoplasmic Securin[IFY-1] levels were measured throughout meiosis I, and the mean ± s.e.m is shown in the graph.

The online version of this article includes the following video and figure supplement(s) for figure 1:

**Figure supplement 1.** PP2A is essential for Meiosis I in *C. elegans* oocytes.

**Figure supplement 2.** CLASP[CLS-2] intensity and localisation is not affected after let-92 depletion.

**Figure 1—video 1.** Wild type and *let-92(RNAi)* oocytes expressing GFP::tubulin and mCherry::histone were dissected and recorded.
https://elifesciences.org/articles/65307#fig1video1

**Figure 1—video 2.** Wild type and *let-92(RNAi)* oocytes expressing GFP::ASPM-1 and mCherry::histone were dissected and recorded.
https://elifesciences.org/articles/65307#fig1video2

**Figure 1—video 3.** Wild type and *let-92(RNAi)* oocytes expressing Securin[IFY-1]::GFP and mCherry::histone were dissected and recorded.
https://elifesciences.org/articles/65307#fig1video3

**Figure 1—video 4.** Wild type and *let-92(RNAi)* oocytes expressing CLASP[CLS-2]::GFP and mCherry::histone were dissected and recorded.
https://elifesciences.org/articles/65307#fig1video4

exact test). Similar defects were observed by depleting the sole PP2A scaffolding subunit, PAA-1 (*Figure 1—figure supplement 1A–C*). In *let-92(RNAi)* and *paa-1(RNAi)*, oocytes microtubules do not organise into a bipolar structure (*Figure 1C*, *Figure 1—figure supplement 1A*, magenta arrows). We then analysed the localisation of the spindle pole protein GFP::ASPM-1 (*Connolly et al., 2014*). GFP::ASPM-1 displays two focused poles in 100% of wild-type oocytes (*Figure 1F,G*, cyan arrows). In contrast, GFP::ASPM-1 foci fail to coalesce in 95% of *let-92(RNAi)* oocytes, displaying a cluster of small foci (*Figure 1F*, orange arrows; *Figure 1G*; *Figure 1—video 2*).

We then sought to determine whether the lack of chromosome segregation was due to a defect in meiosis I progression. We measured endogenously Securin[IFY-1]::GFP levels as a readout for aAnaphase- pPromoting cComplex/cCyclosome (APC/C) activity (i.e. anaphase progression). Cytoplasmic securin[IFY-1] degradation proceeded at similar rates in wild-type and LET-92-depleted oocytes (*Figure 1H,I*; *Figure 1—video 3*), indicating that APC/C activity is largely unperturbed in the absence of PP2A. Given the similarity of this phenotype to the depletion of CLASP orthologue CLS-2 (*Dumont et al., 2010*; *Laband et al., 2017*; *Schlientz and Bowerman, 2020*), we analysed the localisation of CLS-2 and detected no changes in CLS-2 levels throughout meiosis (*Figure 1—figure supplement 2*; *Figure 1—video 4*).

Taken together, we have found that PP2A is required for spindle assembly, chromosome alignment, and PBE during meiosis I. Additionally, APC activity is unperturbed by PP2A depletion, but we cannot rule out an impact on other pathways acting redundantly with APC (*Sonneville and Gonczy, 2004*; *Wang et al., 2013*).

## PP2A:B56 localises to meiotic chromosomes and spindle

To assess the localisation of the core PP2A enzyme (A and C subunits) in vivo, we attempted to tag endogenous LET-92 with GFP but were unsuccessful, presumably due to disruption of its native structure, in agreement with a recent report (*Magescas et al., 2019*). We generated a GFP-tagged version of the sole *C. elegans* PP2A scaffold subunit, PAA-1, and used this to assess the localisation of the core enzyme during meiosis I (*Figure 2A*; *Figure 2—video 1*). The fusion protein localised in centrosomes and P-granules during the early embryonic mitotic divisions (*Figure 2—figure supplement 1A*), in agreement with previous immunofluorescence data (*Lange et al., 2013*). During meiosis I, GFP::PAA-1 localises to spindle poles (*Figure 2A*, cyan arrows) and chromosomes (*Figure 2A*, magenta arrows). The chromosomal signal is a combination of midbivalent and kinetochore populations, visible when analysing single Z-planes in the time-lapses (*Figure 2—figure supplement 1A*, magenta and yellow arrows). As meiosis I progresses, GFP::PAA-1 signal concentrates in the central spindle between the segregating chromosomes (*Figure 2A*, magenta arrowhead), finally disappearing by late anaphase I. This pattern differs substantially from that of Protein Phosphatase 1 (PP1), which localises in the characteristic cup-shaped kinetochores (*Hattersley et al., 2016*).

To address which regulatory subunits are involved in targeting PP2A to the meiotic spindle and chromosomes, we first assessed the localisation of the two *C. elegans* B56 orthologues, PPTR-1 and PPTR-2, and the single B55 orthologue, SUR-6 (*Figure 2—figure supplement 2A*). Live imaging of endogenous, GFP-tagged regulatory subunits revealed that both B56 orthologues, PPTR-1 and PPTR-2, localised strongly and dynamically within the meiotic spindle, whereas no spindle or chromosome localisation of B55[SUR-6] could be detected (*Figure 2—figure supplement 2B*). PPTR-1 has the highest sequence identity with human B56α2 (68.3%) and ε3 (71.87%), whereas PPTR-2 displays the highest sequence identity with human B56δ3 (66.19%) and B56γ1 (68.97%) (*Table 1* and *Supplementary file 1*). Closer analysis of the C-terminal sequence in B56 which plays a key role in specifying centromere versus kinetochore localisation (*Vallardi et al., 2019*) further confirmed that PPTR-1 displays the highest sequence identity with the centromeric B56s, α and ε (93.75% and 87.5%, respectively), while PPTR-2 displays the highest sequence identity with the kinetochore-localised B56, γ and δ (93.75% and 100%, respectively) (*Figure 2—figure supplement 2C*). We will henceforth refer to PPTR-1 as B56α[PPTR-1] and PPTR-2 as B56γ[PPTR-2]. We used endogenous GFP-tagged versions of B56α[PPTR-1] and B56γ[PPTR-2] (*Kim et al., 2017*) and analysed their dynamic localisation pattern in greater spatial and temporal detail. B56α[PPTR-1] localises mainly to the region between homologous chromosomes (midbivalent) during metaphase of meiosis I (*Figure 2B*, magenta arrow; *Figure 2—video 2*) and faintly in kinetochores (*Figure 2B*, yellow arrows). During anaphase I, B56α[PPTR-1] is present exclusively on the central spindle (*Figure 2B*, magenta arrowhead; *Figure 2—video 2*). B56γ[PPTR-2] localises mainly in the midbivalent during metaphase of meiosis I (*Figure 2C*,

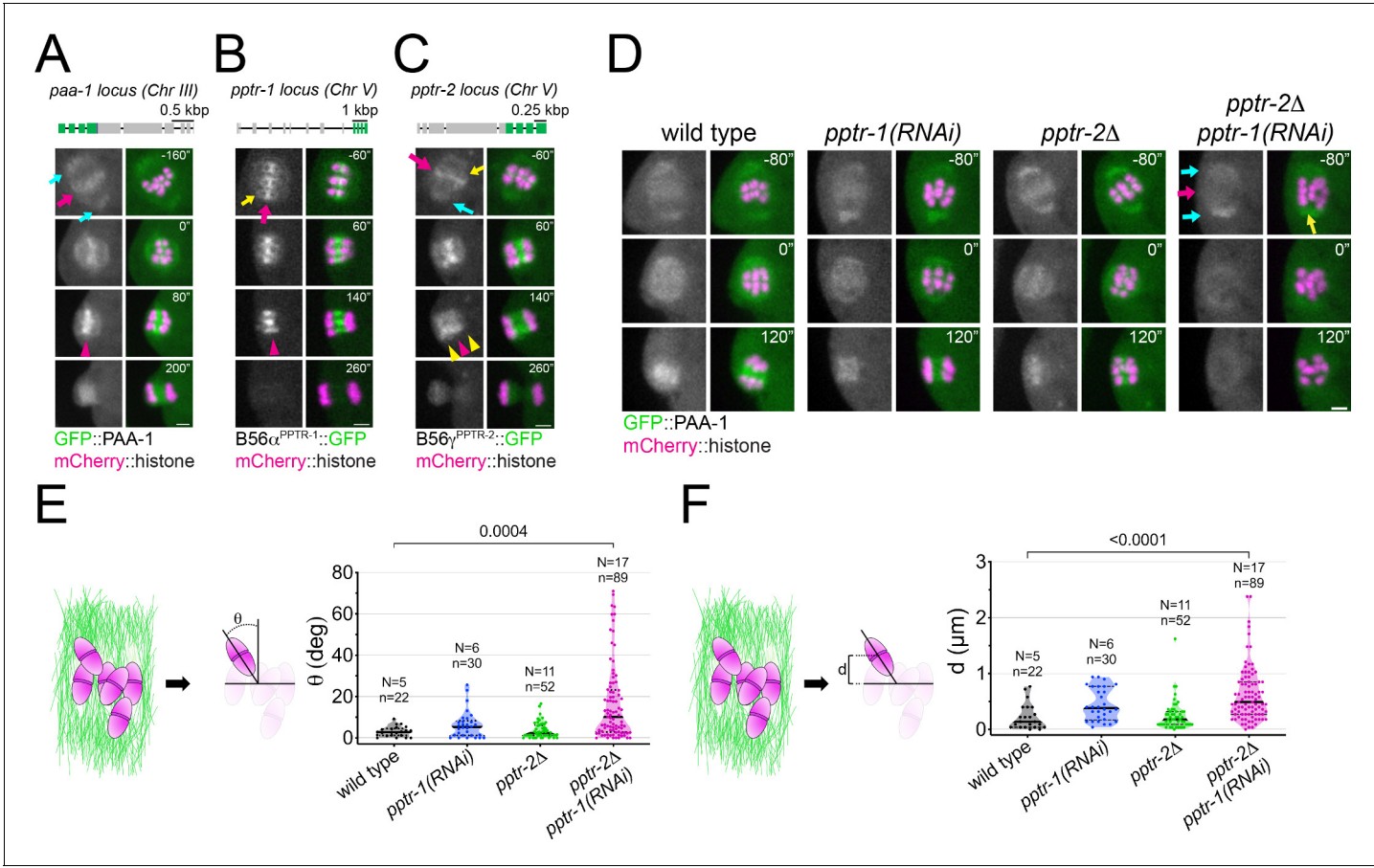

**Figure 2.** *Caenorhabditis elegans* B56 regulatory subunits PPTR-1 and PPTR-2 are required for normal meiosis I. (**A**) Top: Schematic of the *paa-1* gene structure and its tagging with *gfp*. Bottom: The PP2A scaffold subunit GFP::PAA-1 was followed throughout meiosis I in a dissected oocyte. Magenta arrows point to the midbivalent; cyan arrows point to spindle poles; and magenta arrowhead highlights the central-spindle localisation. Inset numbers represent the time relative to metaphase I in seconds. Scale bar, 2 µm. See *Figure 2—video 1*. (**B**) Top: Schematic of the *pptr-1* gene structure and its tagging with *gfp*. Bottom: PPTR-1 and chromosome dynamics were followed in oocytes expressing PPTR-1::GFP and mCherry::histone. The magenta arrow points to the midbivalent and the yellow arrow points to the kinetochore. Magenta arrowhead highlights the central-spindle localisation. Inset numbers represent the time relative to metaphase I in seconds. Scale bar, 2 µm. See *Figure 2—video 2*. (**C**) Top: Schematic of the *pptr-2* gene structure and its tagging with *gfp*. Bottom: PPTR-2 and chromosome dynamics were followed in oocytes expressing PPTR-2::GFP and mCherry::histone. The magenta arrow points to the midbivalent, the yellow arrow points to the kinetochore, and the cyan arrow points to the spindle pole. Inset numbers represent the time relative to metaphase I in seconds. Scale bar, 2 µm. Magenta arrowhead highlights the central-spindle localisation, and yellow arrowhead highlights the chromosome-associated signal. See *Figure 2 —video 3*. (**D**) PAA-1 and chromosome dynamics were followed in wild-type, *pptr-1(RNAi)*, *pptr-2Δ*, and *pptr-2Δ+pptr-1(RNAi)* oocytes expressing GFP::PAA-1 and mCherry::histone. Magenta arrows point to the absence of GFP signal on prometaphase chromosomes, and the cyan arrows point to spindle poles. The yellow arrows highlight one misaligned bivalent. Inset numbers represent the time relative to metaphase I in seconds. Scale bar, 2 µm. See *Figure 2 —video 4*. (**E**) On the left, the schematic depicts how angles were measured relative to the spindle pole-to-pole axis. On the right, the angle of bivalents at 80 s before metaphase I in wild-type, *pptr-1(RNAi)*, *pptr-2Δ*, and *pptr-2Δ+pptr-1(RNAi)* oocytes. Violin plot includes each data point (chromosome), the median (straight black line), and the interquartile range (dashed black lines). N represents number of oocytes and n number of bivalents measured. p values shown in the figure were obtained using a Kruskal–Wallis test. (**F**) On the left, the schematic depicts how distances were measured relative to the centre of the spindle. On the right, the distance of bivalents 80 s before metaphase I was measured in wild-type, *pptr-1(RNAi)*, *pptr-2Δ*, and *pptr-2Δ+pptr-1(RNAi)* oocytes. Violin plot includes each data point (chromosome), the median (straight black line), and the interquartile range (dashed black lines). N represents number of oocytes and n number of bivalents measured. p values shown in the figure were obtained using a Kruskal–Wallis test.

The online version of this article includes the following video and figure supplement(s) for figure 2:

**Figure supplement 1.** Localisation of GFP::PAA-1 during mitosis.

**Figure supplement 2.** Localisation of *C.*

**Figure supplement 3.** Localisation of *C.*

**Figure supplement 4.** Schematics of how congression and alignment were anotated in this study.

**Figure 2—video 1.** An oocyte expressing GFP::PAA-1 and mCherry::histone was dissected and recorded.

https://elifesciences.org/articles/65307#fig2video1

*Figure 2 continued on next page*

*Figure 2 continued*
**Figure 2—video 2.** An oocyte expressing PPTR-1::GFP and mCherry::histone was dissected and recorded.
https://elifesciences.org/articles/65307#fig2video2
**Figure 2—video 3.** An oocyte expressing PPTR-2::GFP and mCherry::histone was dissected and recorded.
https://elifesciences.org/articles/65307#fig2video3
**Figure 2—video 4.** Wild type, *pptr-1(RNAi)*, *pptr-2Δ*, and *pptr-2Δ+pptr-1(RNAi)* oocytes expressing GFP::PAA-1 and mCherry::histone were dissected and recorded.
https://elifesciences.org/articles/65307#fig2video4

magenta arrow; *Figure 2—video 3*), in kinetochores (*Figure 2C*, yellow arrow; *Figure 2—video 3*), and in spindle poles (*Figure 2C*, cyan arrow; *Figure 2—video 3*). A difference between the two paralogues arises during anaphase: B56γ$^{PPTR2}$ is present in the central spindle and on chromosomes (*Figure 2C*, magenta and yellow arrowheads; *Figure 2—video 3*). In spite of this difference, the levels of B56α$^{PPTR-1}$ and B56γ$^{PPTR-2}$ within the meiotic spindle follow similar dynamics (*Figure 2—figure supplement 3A–C*).

These results show that *C. elegans* B56 subunits, like their mammalian counterparts, display a similar but not identical localisation pattern: while both B56α$^{PPTR-1}$ and B56γ$^{PPTR-2}$ are present in the midbivalent and central spindle, B56γ$^{PPTR-2}$ is also associated with chromosomes during anaphase.

## Depletion of B56α$^{PPTR-1}$ and B56γ$^{PPTR-2}$ leads to chromosome congression defects

To address the role of the *C. elegans* B56 subunits during meiosis I, we combined a B56γ$^{PPTR-2}$ deletion allele (*ok1467*, '*pptr-2Δ*') with RNAi-mediated depletion of B56α$^{PPTR-1}$ ('*pptr-1(RNAi)*'), which we will refer to hereafter as 'PPTR-1/2 depletion'. While there was no significant change in GFP::PAA-1 localisation upon depletion of B56α$^{PPTR-1}$ or deletion of B56γ$^{PPTR-2}$, no PAA-1 signal is detected associated with chromosomes upon double PPTR-1/2 depletion (*Figure 2D*, magenta arrow; *Figure 2—video 4*). Of note, PP2A targeting to poles was still detected (*Figure 2D*, cyan arrows), indicating that B56 subunits target PP2A complex to chromosomes but are not necessary for spindle pole targeting (*Figure 2D*; *Figure 2—video 4*). We noticed that chromosomes failed to align upon PPTR-1/2 depletion (*Figure 2D*, yellow arrow), and in order to quantify this phenotype, we measured the angle between the bivalent long axis and the spindle axis ('θ', *Figure 2F*, *Figure 2—figure supplement 4*) and the distance between the centre of the midbivalent and the metaphase plate, defined as the line in the middle of the spindle ('d', *Figure 2G*, *Figure 2—figure supplement 4*). We chose a time of 80 s prior to metaphase I for the analysis since most chromosomes are aligned by this stage (*Figure 2—figure supplement 4*) in agreement with previous data (*Hattersley et al., 2016*). Both the angle (θ) and the distance (d) increased significantly upon PPTR-1/2 double depletion (*Figure 2F,G*). Taken together, these results indicate that B56 subunits are involved in chromosomal and central spindle targeting PP2A and play a role in chromosome congression prior to anaphase. The PAA-1 localisation and chromosome alignment defects observed upon PPTR-1/2 depletion suggest that the subunits are at least partially redundant.

## Shugoshin$^{SGO-1}$ and BUBR1/Mad3$^{SAN-1}$ are not essential for B56 subunit targeting

We next sought to identify the protein(s) involved in targeting PP2A:B56 to meiotic chromosomes. Two of the most studied proteins involved in B56 targeting are Shugoshin and BubR1. The BubR1 orthologue, Mad3$^{SAN-1}$, is not detected in meiotic chromosomes or spindle (*Figure 3—figure*

**Table 1.** Sequence identity between full-length mammalian B56 isoforms and *C. elegans* orthologues.

|  | PPTR-1 | B56β1 | B56ε3 | B56α2 | PPTR-2 | B56δ3 | B56γ1 |
|---|---|---|---|---|---|---|---|
| PPTR-1 |  | 60.12 | 71.87 | 68.30 | 52.26 | 60.17 | 64.48 |
| PPTR-2 | 52.26 | 56.36 | 63.94 | 60.84 |  | 66.19 | 68.97 |

Table was created using Clustal Omega version 2.1. See **Supplementary file 1**.

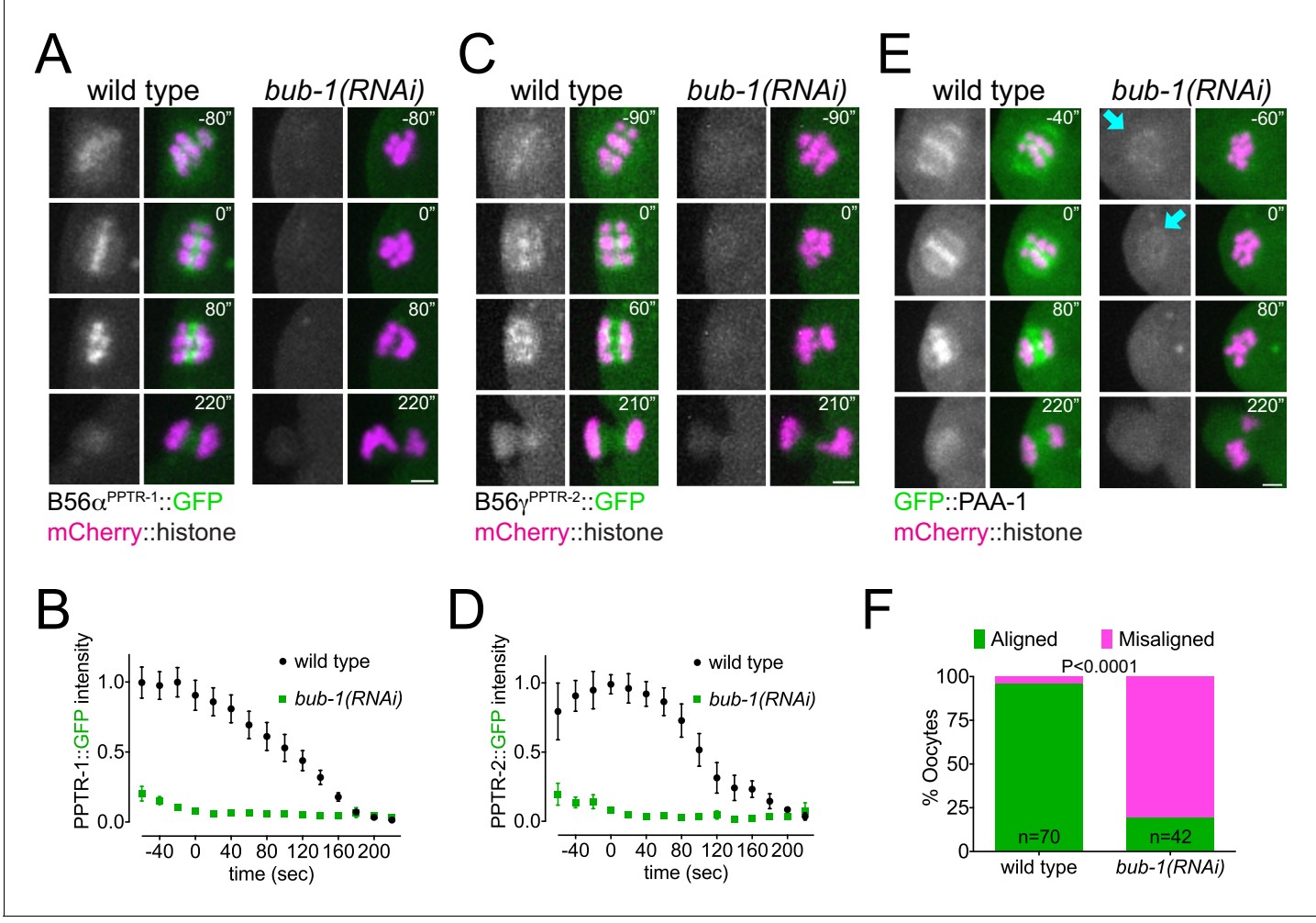

**Figure 3.** BUB-1 recruits B56α$^{PPTR-1}$ and B56γ$^{PPTR-2}$ during oocyte meiosis. (**A**) B56α$^{PPTR-1}$ and chromosome dynamics were followed in wild-type and *bub-1(RNAi)* oocytes expressing PPTR-1::GFP and mCherry::histone. Scale bar, 2 µm. See *Figure 3—video 1*. (**B**) PPTR-1::GFP levels were measured in wild-type and *bub-1(RNAi)* oocytes throughout meiosis I and the mean ± s.e.m is shown in the graph. (**C**) B56γ$^{PPTR-2}$ and chromosome dynamics were followed in wild-type and *bub-1(RNAi)* oocytes expressing PPTR-2::GFP and mCherry::histone. Scale bar, 2 µm. See *Figure 3—video 2*. (**D**) PPTR-2::GFP levels were measured in wild-type and *bub-1(RNAi)* oocytes throughout meiosis I and the mean ± s.e.m is shown in the graph. (**E**) Scaffold subunit PAA-1 and chromosome dynamics were followed in wild-type and *bub-1(RNAi)* oocytes expressing GFP::PAA-1 and mCherry::histone. Cyan arrows highlight the GFP::PAA-1 remaining in *bub-1(RNAi)* oocytes. Scale bar, 2 µm. See *Figure 3—video 3*. (**F**) The number of oocytes with misaligned chromosomes at metaphase I in wild-type and *bub-1(RNAi)* oocytes was analysed, and the percentage is represented (p<0.0001, Fisher's exact test).

The online version of this article includes the following video and figure supplement(s) for figure 3:

**Figure supplement 1.** Mad3$^{SAN-1}$ does not play a major role in B56α$^{PPTR-1}$ and B56γ$^{PPTR-2}$ targeting.

**Figure supplement 2.** Shugoshin$^{SGO-1}$ does not play a major role in B56α$^{PPTR-1}$ and B56γ$^{PPTR-2}$ targeting.

**Figure supplement 3.** PP1 catalytic subunit localisation is not regulated by BUB-1.

**Figure 3—video 1.** Wild type and *bub-1(RNAi)* oocytes expressing PPTR-1::GFP and mCherry::histone were dissected and recorded.

https://elifesciences.org/articles/65307#fig3video1

**Figure 3—video 2.** Wild type and *bub-1(RNAi)* oocytes expressing PPTR-2::GFP and mCherry::histone were dissected and recorded.

https://elifesciences.org/articles/65307#fig3video2

**Figure 3—video 3.** Wild type and *bub-1(RNAi)* oocytes expressing GFP::PAA-1 and mCherry::histone were dissected and recorded.

https://elifesciences.org/articles/65307#fig3video3

**Figure 3—video 4.** Wild type and *bub-1(RNAi)* oocytes expressing GFP::PP1c$^{GSP-2}$ and mCherry::histone were dissected and recorded.

https://elifesciences.org/articles/65307#fig3video4

supplement 1A) and Mad3$^{SAN-1}$ depletion by RNAi or a Mad3$^{SAN-1}$ deletion allele (*ok1580*) does not inhibit B56 localisation (*Figure 3—figure supplement 1B,C*). We therefore tested the role of the

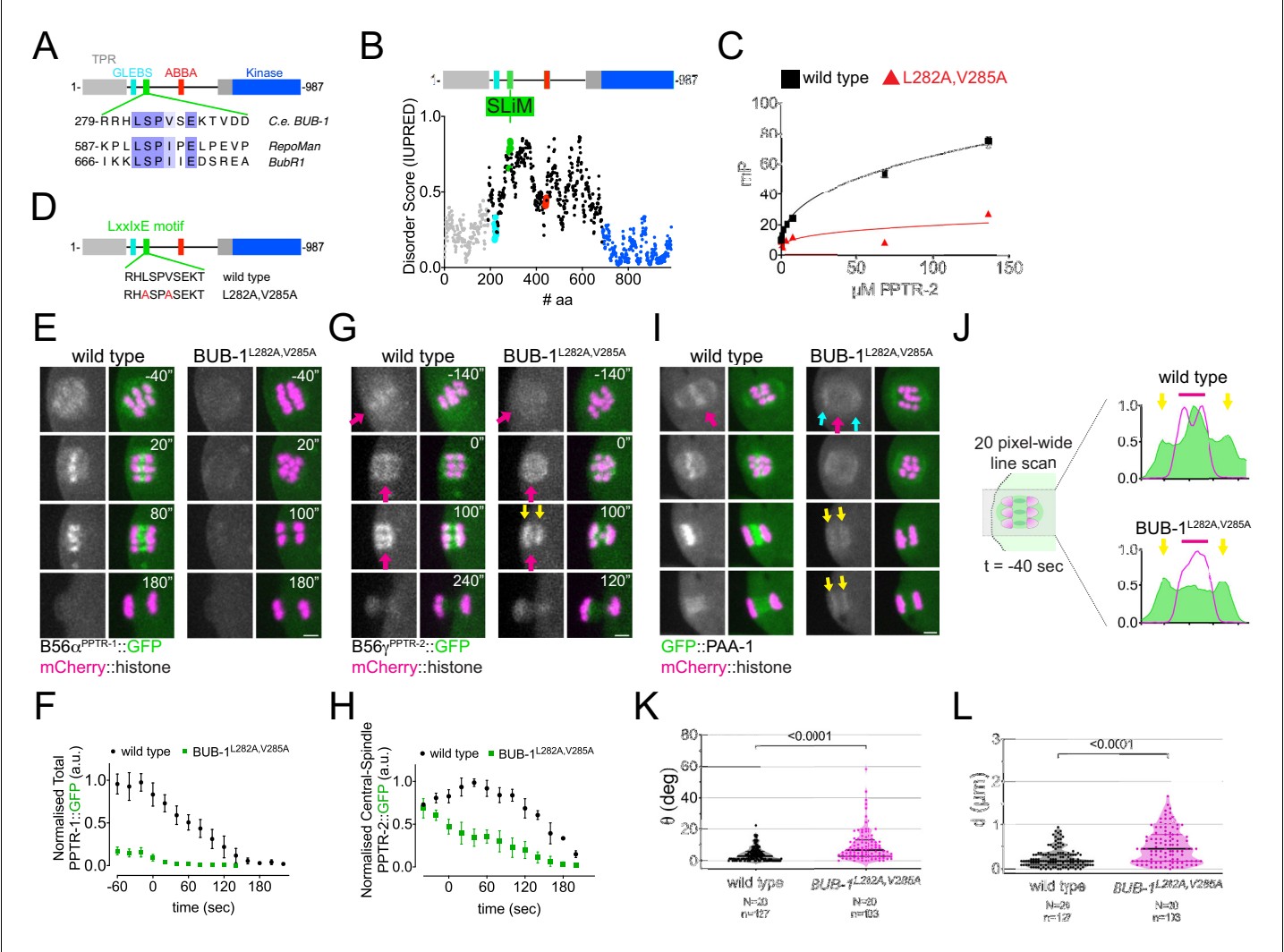

**Figure 4.** B56α^PPTR-1 and B56γ^PPTR-2 are targeted through a LxxIxE motif in BUB-1. (**A**) *C. elegans* BUB-1 LxxIxE SLiM is shown aligned with BubR1 and Repoman short linear motifs (SLiMs). The only change within the LxxIxE sequence itself is the presence of a valine instead of isoleucine, which still fits within the consensus (***Wang et al., 2016***). The alignment was performed using Clustal Omega and visualised with Jalview (***Waterhouse et al., 2009***). (**B**) Disorder prediction of full-length BUB-1 was done using IUPRED2A (***Erdős and Dosztányi, 2020***). All prevuously characterised BUB-1 domains fall within ordered regions < 0.5. The putative SLiM is in a disordered region (IUPRED score ~0.8). (**C**) PPTR-2 interaction with a LxxIxE motif-containing synthetic peptide was assessed using fluorescence polarisation. Increasing amounts of purified recombinant PPTR-2 were incubated with FITC-labelled wild-type or L282A,V285A mutant peptide. The graph was taken from a representative experiment and shows the mean ± s.d. of technical triplicates. (**D**) Schematic showing the LxxIxE motif in BUB-1 and the BUB-1^L282A,V285A mutant. (**E**) B56α^PPTR-1 and chromosome dynamics were followed in wild-type and in BUB-1^L282A,V285A oocytes expressing PPTR-1::GFP and mCherry::histone. Scale bar, 2 μm. Inset numbers represent the time relative to metaphase I in seconds. See ***Figure 4—video 1***. (**F**) PPTR-1::GFP levels were measured in wild-type and BUB-1^L282A,V285A oocytes throughout meiosis I and the mean ± s.e.m is shown in the graph. (**G**) B56γ^PPTR-2 and chromosome dynamics were followed in wild-type and in BUB-1^L282A,V285A oocytes expressing PPTR-2::GFP and mCherry::histone. Magenta arrows point towards prometaphase chromosomes and anaphase central spindle, whereas yellow arrows point towards the chromosome-associated anaphase signal. Scale bar, 2 μm. Inset numbers represent the time relative to metaphase I in seconds. See ***Figure 4—video 2***. (**H**) Midbivalent/central spindle PPTR-2::GFP levels were measured in wild-type and BUB-1^L282A,V285A oocytes throughout meiosis I, and the mean ± s.e.m is shown in the graph. (**I**) GFP::PAA-1 is present on chromosomes in the BUB-1^L282A,V285A mutant. PAA-1 and chromosome dynamics were followed in wild-type and BUB-1^L282A,V285A oocytes expressing GFP::PAA-1 and mCherry::histone. Magenta arrows point towards prometaphase chromosomes and anaphase central spindle, whereas yellow arrows point towards the chromosome-associated anaphase signal. Cyan arrows point towards spindle poles. Scale bar, 2 μm. Inset numbers represent the time relative to metaphase I in seconds. See ***Figure 4—video 3***. (**J**) Representative, spindle-wide (20 pixels) line profiles are shown for wild-type (top) and BUB-1^L282A,V285A mutant (bottom) measured in prometaphase I (40 s before metaphase I). Green arrows point to the PAA-1 pole signal and blue line to the chromosome associated population. (**K**) The angle of bivalents 80 s before segregation in meiosis I relative to the average angle of the spindle was measured in wild-type and in BUB-1^L282A,V285A oocytes. Violin plot includes each data point (chromosome), the median (straight black line), and the interquartile range (dashed black lines). N represents

*Figure 4 continued on next page*

*Figure 4 continued*

number of oocytes and n number of bivalents measured. p value shown in the figure was obtained using a Mann–Whitney test. (**L**) Distance of bivalents 80 s before segregation in meiosis I relative to the centre of the spindle was measured in wild-type and BUB-1[L282A,V285A] oocytes. Violin plot includes each data point (chromosome), the median (straight black line), and the interquartile range (dashed black lines). N represents number of oocytes and n number of bivalents measured. p value shown in the figure was obtained using a Mann–Whitney test.

The online version of this article includes the following video and figure supplement(s) for figure 4:

**Figure supplement 1.** Alignment of the B56 subunits LxxIxE motif binding pocket.

**Figure supplement 2.** Mad1[MDF-1] localisation is not affected by a LxxIxE motif mutation in BUB-1.

**Figure 4—video 1.** Wild type and *BUB-1[L282A,V285A]* oocytes expressing PPTR-1::GFP and mCherry::histone were dissected and recorded.
https://elifesciences.org/articles/65307#fig4video1

**Figure 4—video 2.** Wild type and *BUB-1[L282A,V285A]* oocytes expressing PPTR-2::GFP and mCherry::histone were dissected and recorded.
https://elifesciences.org/articles/65307#fig4video2

**Figure 4—video 3.** Wild type and *BUB-1[L282A,V285A]* oocytes expressing GFP::PAA-1 and mCherry::histone were dissected and recorded.
https://elifesciences.org/articles/65307#fig4video3

Shugoshin orthologue, SGO-1. We used a *sgo-1* deletion allele generated by CRISPR (*Ferrandiz et al., 2018*; *Figure 3—figure supplement 1*) and observed no change in B56α[PPTR-1] levels (*Figure 3—figure supplement 2B,C*) and a slight reduction in B56γ[PPTR-2] levels (*Figure 3—figure supplement 2D,E*). Particularly during metaphase I, PPTR-2 intensity only decreased between 20 and 40% in the *sgo-1Δ* mutant (*Figure 3—figure supplement 2*, shaded area). Hence, Mad3[SAN-1] and Shugoshin[SGO-1] are not essential in PP2A:B56 targeting during meiosis in *C. elegans* oocytes.

## BUB-1 targets B56 subunits through a conserved LxxIxE motif

*C. elegans* B56 subunits display a dynamic localisation pattern similar to that of the kinase BUB-1 throughout meiosis I (*Dumont et al., 2010*; *Monen et al., 2005*; *Pelisch et al., 2019*; *Pelisch et al., 2017*), and interestingly, BUB-1 depletion by RNAi or using an auxin-inducible degradation system leads to alignment and segregation defects during meiosis (*Dumont et al., 2010*; *Pelisch et al., 2019*). We therefore tested whether BUB-1 is involved in the recruitment of B56 subunits to meiotic chromosomes. RNAi-mediated depletion of BUB-1 abolished meiotic chromosome localisation of B56α[PPTR-1] (*Figure 3A,B*; *Figure 3—video 1*) and B56γ[PPTR-2] (*Figure 3C,D*; *Figure 3—video 2*). GFP::PAA-1 localisation on chromosomes was also abolished by BUB-1 depletion (*Figure 3E*; *Figure 3—video 3*). While we could not address the intensity on spindle poles due to the strong spindle defect in the absence of BUB-1, we could detect GFP::PAA-1 in extrachromosomal regions (*Figure 3E*, cyan arrows), indicating that BUB-1 specifically regulates PP2A:B56 chromosomal targeting. Consistent with this idea, BUB-1 depletion does not affect the localisation of the catalytic subunit of another major phosphatase, PP1[GSP-2] (*Figure 3—figure supplement 3*; *Figure 3—video 4*). In agreement with published data (*Dumont et al., 2010*), BUB-1 depletion causes severe defects in chromosome alignment (*Figure 3F*).

*C. elegans* BUB-1 contains a conserved N-terminal tetratricopeptide repeat domain, a C-terminal kinase domain, and regions mediating its interaction with Cdc20 and BUB3 (ABBA and GLEBS motifs, respectively) (*Figure 4A*; *Kim et al., 2017*; *Kim et al., 2015*; *Moyle et al., 2014*). Sequence analysis of *C. elegans* BUB-1 revealed the presence of a putative B56 LxxIxE motif in residues 282–287 (*Figure 4A*). When compared with two well-characterised LxxIxE motifs, in BubR1 and Repo-Man, there is a high degree of conservation in the key residues making contacts with the B56 hydrophobic pockets (*Figure 4A*). In addition, the SLiM-binding hydrophobic pocket in B56 subunits is very well conserved in *C. elegans* (*Figure 4—figure supplement 1*). The putative LxxIxE motif lies within a region predicted to be disordered (*Figure 4B*), with serine 283 fitting a Cdk1 Ser/Thr-Pro motif.

Using a fluorescence polarisation-based assay, we confirmed that the LxxIxE motif of BUB-1 binds to purified recombinant PPTR-2, and this binding is abolished by the L282A,V285A mutations (*Figure 4C*). While localisation of BUB-1[L282A,V285A] was indistinguishable from that of wild-type BUB-1 during metaphase I (*Figure 4—figure supplement 2A*), localisation of B56α[PPTR-1] to the midbivalent and central spindle was almost completely lost in the BUB-1[L282A,V285A] mutant (*Figure 4D–F*; *Figure 4—video 1*). B56γ[PPTR-2] midbivalent localisation leading to metaphase I and central-spindle localisation during anaphase were also dependent on the BUB-1 LxxIxE motif (*Figure 4G*, magenta

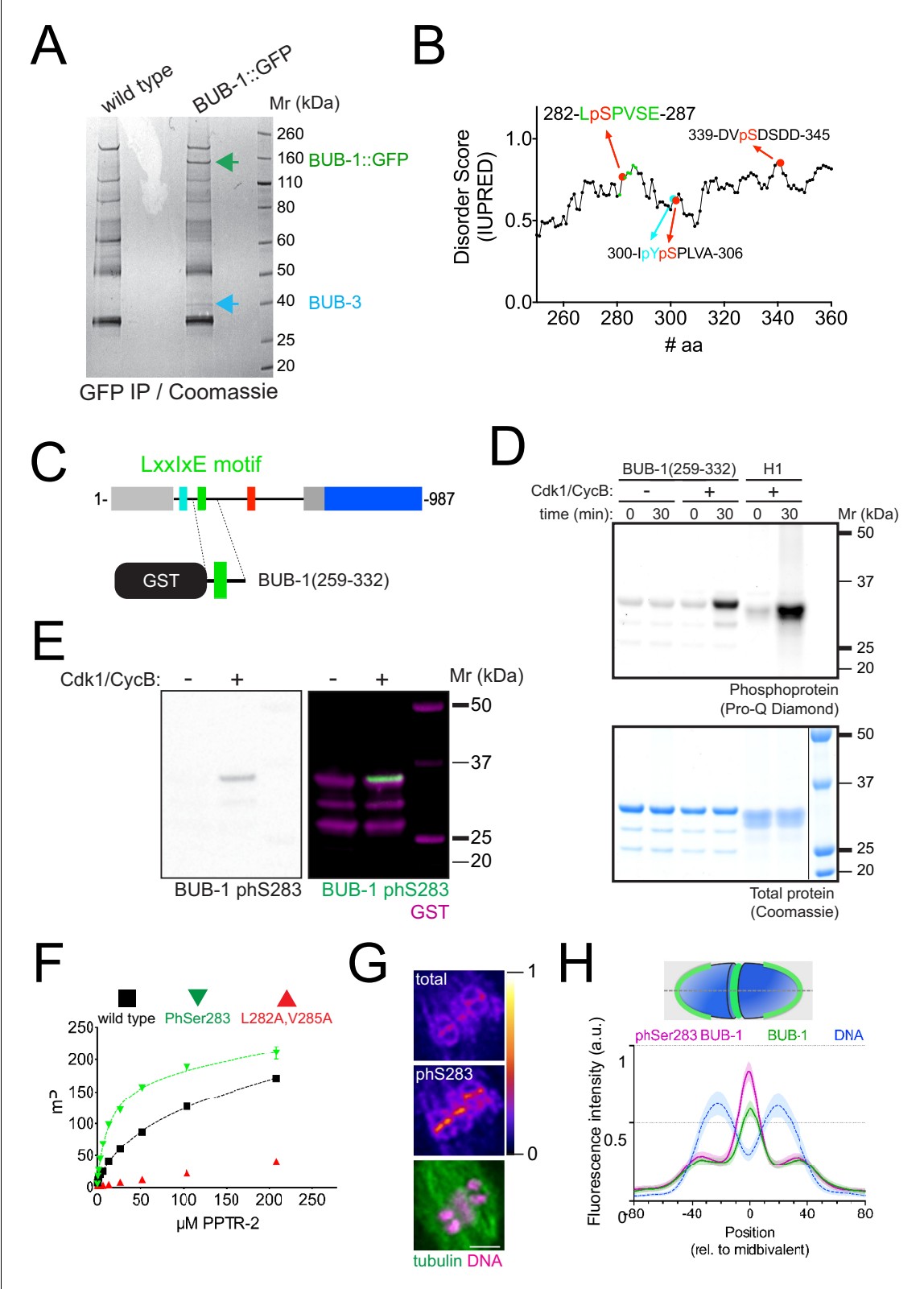

**Figure 5.** Phosphorylation of Ser 283 within the LxxIxE motif regulates B56 subunit binding. (A) BUB-1::GFP was immunoprecipitated and the eluted proteins were run on sodium dodecyl sulfate–polyacrylamide gel electrophoresis (SDS–PAGE) followed by Coomassie staining. The position of BUB-1:: GFP and BUB-3 is highlighted with green and blue arrows, respectively. The band corresponding to BUB-1::GFP was further analysed by mass spectrometry to look for phospho-modified peptides. (B) Serine 283 is phosphorylated in vivo. In addition, three other phosphorylation sites were

*Figure 5 continued on next page*

Figure 5 continued

identified within this disordered region. (C) A fragment of BUB-1 (259–332) containing the LxxIxE motif (282–287) was expressed in bacteria fused to a GST tag. (D) GST-BUB-1 (259–332) was phosphorylated in vitro using Cdk1/CyclinB ('Cdk1/CycB'). Histone H1 was used as a positive control. Reactions were run on 4–12% SDS–PAGE and subject to phosphoprotein staining (top) followed by total protein staining (bottom). (E) GST-BUB-1 (259–332) was phosphorylated in vitro using Cdk1/CyclinB ('Cdk1/CycB') and subject to western blotting using a specific antibody against phosphorylated serine 283. Anti-GST served as a loading control for the substrate. (F) PPTR-2 interaction with a LxxIxE motif-containing synthetic peptide was assessed using fluorescence polarisation. Increasing amounts of purified recombinant PPTR-2 were incubated with FITC-labelled wild-type, serine 283 phosphorylated, or L282A,V285A mutant peptide. The graph was taken from a representative experiment and shows the mean ± s.d. of technical triplicates. (G) Fixed oocytes were subject to immunofluorescence using labelled BUB-1 ('total'), phospho Ser 283-BUB-1 ('phS283'), and tubulin. Single-channel images for BUB-1 and phospho Ser 283-BUB-1 are shown in 'fire' LUT, and the bottom panel shows tubulin (green) and DNA (magenta). (H) Line profile analysis was performed in samples co-stained with labelled BUB-1 (Alexa488) and phospho Ser 283-BUB-1 (Alexa647) as described in the Materials and methods section. The lines represent the mean, and the shaded area represents the s.d. of the indicated number of bivalents.

The online version of this article includes the following figure supplement(s) for figure 5:

**Figure supplement 1.** BUB-1::GFP immunoprecipitation and selected MS spectra of peptides containing phospho serine 283.
**Figure supplement 2.** Validation of anti-phospho Ser 283 BUB-1 specific antibody.

---

arrows, and *Figure 4H*; *Figure 4—video 2*). In contrast, chromosome-associated B56γ$^{PPTR-2}$ was less affected in the BUB-1$^{L282A,V285A}$ mutant (*Figure 4G*, yellow arrows). Importantly, the LxxIxE motif mutations BUB-1$^{L282A,V285A}$ abrogates GFP::PAA-1 localisation in the midbivalent and central spindle (*Figure 4I*, magenta arrows; *Figure 4—video 3*) without affecting the spindle pole PAA-1 localisation (*Figure 4I*, cyan arrows; *Figure 4—video 3*). Some GFP::PAA-1 signal was still detected in anaphase chromosomes, consistent with the remaining B56γ$^{PPTR-2}$ on chromosomes (*Figure 4I,J*, yellow arrows). This LxxIxE motif-dependent regulation is specific for B56 subunits because the localisation of the SAC component Mad1$^{MDF-1}$, a known BUB-1 interactor (*Moyle et al., 2014*), is not disrupted after mutating the BUB-1 LxxIxE motif (*Figure 4—figure supplement 2B*). Mutation of the BUB-1 LxxIxE motif not only impairs B56 chromosomal targeting but also leads to alignment/congression defects (*Figure 4K,L*), indicating that the BUB-1 LxxIxE motif recruits PP2A:B56 and plays an important role in chromosome congression. Lagging chromosomes were detected in 36% of oocytes in the BUB-1$^{L282A,V285A}$ mutant, versus 10% in wild-type oocytes (p=0.0125, Fisher's exact test), but segregation was achieved in 100% of the cases with no PBE defects were detected. Interestingly, the newly identified LxxIxE motif specifically recruits PP2A:B56 to metaphase chromosomes and anaphase central spindle, but not anaphase chromosomes.

## Phosphorylation of BUB-1 LxxIxE is important for PP2A:B56 chromosome targeting

Since phosphorylation of LxxIxE motifs can increase affinity for B56 by ~10 fold (*Wang et al., 2016*), we sought to determine whether Ser 283 within BUB-1 LxxIxE motif is phosphorylated in vivo. To this end, we immunoprecipitated endogenous, GFP-tagged BUB-1 from embryo lysates using a GFP nanobody. Immunoprecipitated material was digested with trypsin, peptides analysed by mass spectrometry and searches conducted for peptides containing phosphorylated adducts. As expected GFP-tagged, but not untagged, BUB-1 was pulled down by the GFP nanobody (*Figure 5—figure supplement 1A*) along with the BUB-1 partner, BUB-3 (*Figure 5A*, blue arrow). We found four phospho-sites clustered within a disordered region, one of which is Ser 283, within the LxxIxE motif (*Figure 5B*; see also representative MS spectra for Ser 283 in *Figure 5—figure supplement 1B*). We expressed a BUB-1 fragment containing the LxxIxE motif (*Figure 5C*) and performed Cdk1 kinase assays, which showed that Cdk1 can phosphorylate the 259–332 BUB-1 fragment (*Figure 5D*). Western blot analysis using a phospho-specific antibody revealed that Cdk1 can phosphorylate Ser 283 in vitro (*Figure 5E*). Importantly, LxxIxE motif peptides with phosphorylated Ser 283 bound recombinant B56γ$^{PPTR-2}$ with higher affinity than non-phosphorylated LxxIxE peptide (*Figure 5F*). We generated a phospho Ser 283-specific BUB-1 antibody (*Figure 5—figure supplement 2*) and observed that while BUB-1 phosphorylated in Ser 283 is detected in midbivalent and kinetochore, it is preferentially enriched in the midbivalent (*Figure 5G,H*).

Mutating the Ser 283 to Ala in endogenous BUB-1 ('BUB-1$^{S283A}$') had a similar effect to that of the BUB-1$^{L282A,V285A}$ mutant: it significantly reduced B56α$^{PPTR-1}$ localisation (*Figure 6A,B*; *Figure 6—video 1*), and in the case of B56γ$^{PPTR-2}$, it significantly reduced its midbivalent and central spindle

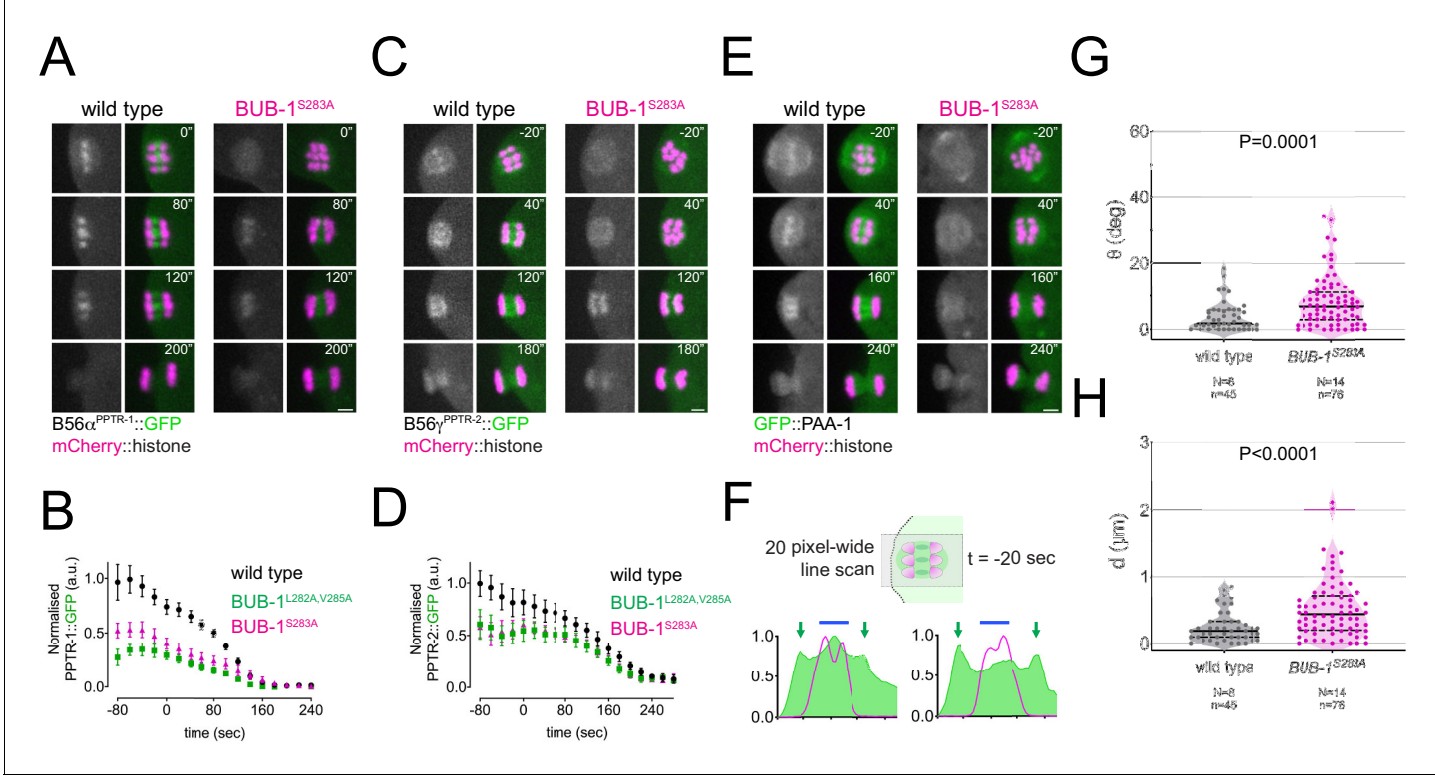

**Figure 6.** LxxIxE motif phosphorylation regulates the recruitment of B56s subunits in vivo. (**A**) B56α$^{PPTR-1}$ and chromosome dynamics were followed in wild-type and in BUB-1$^{S283A}$ oocytes expressing PPTR-1::GFP and mCherry::histone. Scale bar, 2 μm. Inset numbers represent the time relative to metaphase I in seconds. See *Figure 6—video 1*. (**B**) PPTR-1::GFP levels were measured in wild-type, BUB-1$^{L282A,V285A}$ and in BUB-1$^{S283A}$ oocytes throughout meiosis I and the mean ± s.e.m. is shown in the graph. (**C**) B56γ$^{PPTR-2}$ and chromosome dynamics were followed in wild-type and in BUB-1$^{S283A}$ oocytes expressing PPTR-2::GFP and mCherry::histone. Scale bar, 2 μm. Inset numbers represent the time relative to metaphase I in seconds. See *Figure 6—video 2*. (**D**) PPTR-2::GFP levels were measured in wild-type, BUB-1$^{L282A,V285A}$, and BUB-1$^{S283A}$ oocytes throughout meiosis I, and the mean ± s.e.m is shown in the graph. (**E**) GFP::PAA-1 is present on chromosomes in the BUB-1$^{S283A}$ mutant. PAA-1 and chromosome dynamics were followed in wild-type and in BUB-1$^{S283A}$ oocytes expressing GFP::PAA-1 and mCherry::histone. Yellow dotted line shows that chromosome associated PAA-1 is lost in the BUB-1$^{S283A}$ mutant. Scale bar, 2 μm. Inset numbers represent the time relative to metaphase I in seconds. See *Figure 6—video 3*. (**F**) Representative, spindle-wide (20 pixels) line profiles are shown for wild-type (left) and BUB-1$^{S283A}$ mutant (right) measured in prometaphase I (20 s before metaphase I). Green arrows point to the PAA-1 pole signal and blue line to the chromosome associated population. (**G**) Angle of bivalents 80 s before segregation in meiosis I relative to the average angle of the spindle was measured in wild-type and in BUB-1$^{S283A}$ oocytes. Violin plot includes each data point (chromosome), the median (straight black line), and the interquartile range (dashed black lines). N represents number of oocytes and n number of bivalents measured. p value shown in the figure was obtained using a Mann–Whitney test. (**H**) Distance of bivalents 80 s before segregation in meiosis I relative to the centre of the spindle was measured in wild-type and in BUB-1$^{S283A}$ oocytes. Violin plot includes each data point (chromosome), the median (straight black line), and the interquartile range (dashed black lines). N represents number of oocytes and n number of bivalents measured. p value shown in the figure was obtained using a Mann–Whitney test.

The online version of this article includes the following video(s) for figure 6:

**Figure 6—video 1.** Wild type and *BUB-1$^{S283A}$* oocytes expressing PPTR-1::GFP and mCherry::histone were dissected and recorded.
https://elifesciences.org/articles/65307#fig6video1
**Figure 6—video 2.** Wild type and *BUB-1$^{S283A}$* oocytes expressing PPTR-2::GFP and mCherry::histone were dissected and recorded.
https://elifesciences.org/articles/65307#fig6video2
**Figure 6—video 3.** Wild type and *BUB-1$^{S283A}$* oocytes expressing GFP::PAA-1 and mCherry::histone were dissected and recorded.
https://elifesciences.org/articles/65307#fig6video3

localisation (*Figure 6C,D*; *Figure 6—video 2*). In line with these results and consistent with a B56-dependent chromatin recruitment of PP2A, GFP::PAA-1 was not detected on chromosomes during prometaphase/metaphase in the BUB-1$^{S283A}$ mutant (*Figure 6E,F*; *Figure 6—video 3*). BUB-1$^{S283A}$ mutation not only impairs PP2A:B56 chromosomal targeting but also leads to alignment/congression defects (*Figure 6G,H*), indicating that phosphorylation of Ser 283 within the BUB-1 LxxIxE motif recruits PP2A:B56 and plays an important role in chromosome congression.

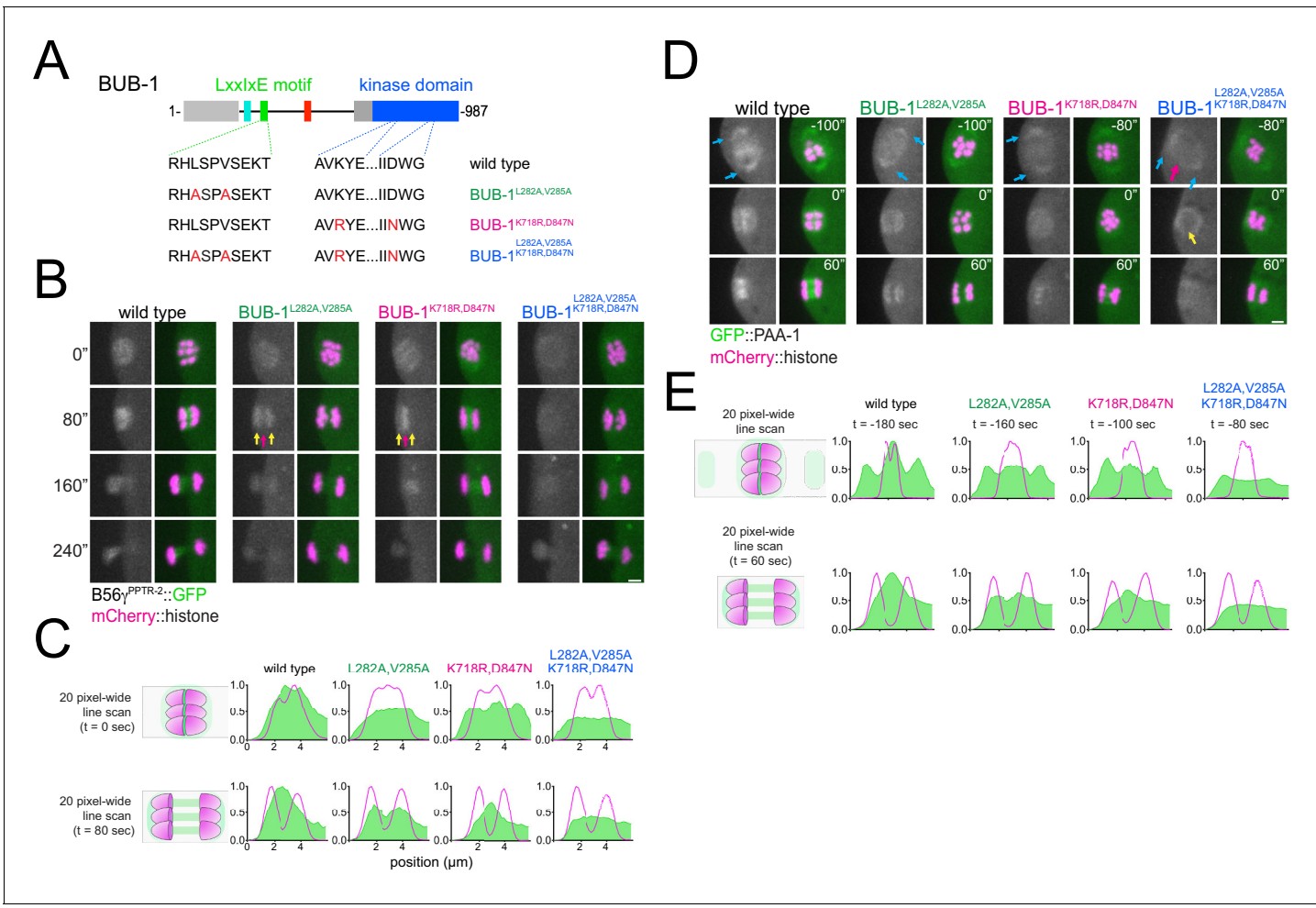

**Figure 7.** Role of BUB-1 kinase domain in B56γ^PPTR-2 chromosomal targeting. (**A**) Schematic showing the LxxIxE motif and the kinase domain in BUB-1 of the wild-type, the BUB-1^L282A,V285A, BUB-1^K718R,D847N, and BUB-1^L282A,V285A, K718R,D847N mutant. (**B**) B56γ^PPTR-2 and chromosome dynamics were followed in wild-type, the BUB-1^L282A,V285A, BUB-1^K718R,D847N, and BUB-1^L282A,V285A, K718R,D847N oocytes expressing PPTR-2::GFP and mCherry::histone. Magenta arrows point towards the central spindle and yellow arrows towards chromosomes. Scale bar, 2 μm. Inset numbers represent the time relative to metaphase I in seconds. See also *Figure 7—video 2*. (**C**) Representative, spindle-wide (20 pixels) line profiles are shown for wild-type, the BUB-1^L282A,V285A, BUB-1^K718R,D847N, and BUB-1^L282A,V285A, K718R,D847N. On top profiles of metaphase plate just in anaphase onset, on the bottom profiles 80 s after segregation to show central spindle levels. (**D**) Scaffolding subunit PAA-1 and chromosome dynamics were followed in wild-type, the BUB-1^L282A, V285A, BUB-1^K718R,D847N, and BUB-1^L282A,V285A, K718R,D847N oocytes expressing GFP::PAA-1 and mCherry::histone. Cyan arrows point to pole population of PAA-1, magenta arrow signals chromosomes, and yellow arrow highlight the levels of PAA-1 in anaphase onset. Scale bar, 2 μm. Inset numbers represent the time relative to metaphase I in seconds. See also *Figure 7—video 3*. (**E**) Representative, spindle-wide (20 pixels) line profiles are shown for wild-type, the BUB-1^L282A,V285A, BUB-1^K718R,D847N, and BUB-1^L282A,V285A, K718R,D847N. On top profiles before segregation starts (different time points matching the first panel on D) to highlight the PAA-1 in the poles and on the bottom profiles 60 s after segregation to show central spindle levels.

The online version of this article includes the following video and figure supplement(s) for figure 7:

**Figure supplement 1.** Kinase domain in BUB-1 does not regulate targeting of B56α^PPTR-1 regulatory subunit in meiosis I.

**Figure supplement 2.** Embryo viability and brood size analysis of BUB-1 mutants.

**Figure 7—video 1.** Wild type and *BUB-1^K718R,D847N* oocytes expressing PPTR-1::GFP and mCherry::histone were dissected and recorded.

https://elifesciences.org/articles/65307#fig7video1

**Figure 7—video 2.** Wild type and *BUB-1^L282A,V285A*, *BUB-1^K718R,D847N*, *BUB-1 ^L282A,V285A;K718R,D847N* oocytes expressing PPTR-2::GFP and mCherry::histone were dissected and recorded.

https://elifesciences.org/articles/65307#fig7video2

**Figure 7—video 3.** Wild type and *BUB-1^L282A,V285A*, *BUB-1^K718R,D847N*, *BUB-1 ^L282A,V285A;K718R,D847N* oocytes expressing GFP::PAA-1 and mCherry::histone were dissected and recorded.

https://elifesciences.org/articles/65307#fig7video3

## Anaphase chromosomal recruitment of PP2A:B56$\gamma^{\text{PPTR-2}}$ depends on BUB-1 C-terminal kinase domain

Since not all of B56$\gamma^{\text{PPTR-2}}$ is targeted by the BUB-1 LxxIxE motif, we turned our attention to BUB-1 kinase domain. In *C. elegans*, BUB-1 kinase domain regulates its interaction with binding partners such as Mad1$^{\text{MDF-1}}$. The K718R/D847N double mutation (equivalent to positions K821 and D917 in human Bub1) destabilises the kinase domain and prevents its interaction with Mad1$^{\text{MDF-1}}$ (*Moyle et al., 2014*). We generated endogenous BUB-1$^{\text{K718R,D847N}}$ (*Figure 7A*) and confirmed that these mutations abolish recruitment of Mad1$^{\text{MDF-1}}$ (*Figure 4—figure supplement 2B*). B56$\alpha^{\text{PPTR-1}}$:: GFP localisation and dynamics remained largely unaltered in the mutant BUB-1$^{\text{K718R,D847N}}$ (*Figure 7—figure supplement 1*; *Figure 7—video 1*). The B56$\gamma^{\text{PPTR-2}}$::GFP pool most affected by the K718R,D847N mutant was the chromosomal one (*Figure 7B*, yellow arrows; *Figure 7—video 2*), with the central-spindle pool reduced but still present (*Figure 7B*, magenta arrow; *Figure 7—video 2*). We therefore combined the K718R,D847N with the L282A,V285A mutations to generate 'BUB-1$^{\text{L282A,V285A;K718R,D847N}}$' observed a significant decrease in both chromosomal and central spindle B56$\gamma^{\text{PPTR-2}}$::GFP (*Figure 7B*; *Figure 7—video 2*). Line profiles highlighting these localisation differences in the mutants are shown in *Figure 7C*. We then analysed GFP::PAA-1 was not found associated with chromosomes during prometaphase I in the BUB-1$^{\text{L282A,V285A;K718R,D847N}}$ mutant (*Figure 7E*, magenta arrow; *Figure 7—video 3*). Of note, a significant population of GFP::PAA-1 remained associated with spindles poles (*Figure 7D*, cyan arrows) and surrounding the spindle area during anaphase (*Figure 7D*, yellow arrows).

These results show that the BUB-1 kinase domain is important for recruitment of B56$\gamma^{\text{PPTR-2}}$, but not of B56$\alpha^{\text{PPTR-1}}$. Thus, BUB-1 is a key factor for the recruitment of PP2A:B56 but could also provide the basis for establishing (probably together with other proteins) the two pools of B56$\gamma^{\text{PPTR-2}}$.

## Discussion

In the present article, we have uncovered new roles for PP2A during oocyte meiosis in *C. elegans*. PP2A is essential for meiosis I: depletion of the catalytic or scaffold subunits leads to severe spindle assembly defects, lack of chromosome segregation, and failure to achieve PBE. These effects are likely brought about by a combination of different PP2A subcomplexes with varying regulatory B subunits. PP2A:B56 regulates chromosome dynamics prior to segregation since depletion of the two B56 orthologues, PPTR-1 and PPTR-2, leads to alignment defects. We have uncovered a new phospho-regulated B56 LxxIxE motif in *C. elegans* BUB-1 that recruits PP2A:B56 and is important for chromosome alignment during meiosis I.

### Dissecting the role of PP2A during oocyte meiosis

Depletion of the sole catalytic or scaffold PP2A subunits leads to massive meiotic failure and embryonic lethality. The earliest effect we could see in our experimental set up is a failure to assemble a bipolar spindle. This will of course have a direct impact on any process that should occur after spindle assembly, including chromosome alignment and segregation, followed by PBE. However, BUB-1 depletion leads to severe spindle assembly and alignment defects, yet in the majority of cases, chromosomes segregate (with visible errors – lagging chromosomes) and polar bodies do extrude. Therefore, lack of a proper bipolar metaphase spindle is not sufficient to result in lack of segregation or PBE. PP2A complexes harbouring the B56 subunits PPTR-1 and PPTR-2 are required to target the phosphatase to chromosomes to regulate chromosome alignment in metaphase I. Furthermore, this chromosomal B56 pool is recruited by BUB-1 through its LxxIxE SLiM motif. Mutation of this motif that prevents binding to B56 subunits leads to alignment defects. B56 subunits however are not involved in spindle pole targeting of PP2A, suggesting that this spindle pole pool is the one relevant for spindle assembly. Our attempts to address a possible role for other B subunits were focused on RSA-1 and SUR-6 given their reported centrosomal roles during mitosis (*Enos et al., 2018*; *Kitagawa et al., 2011*; *Schlaitz et al., 2007*; *Song et al., 2011*). However, neither RSA-1 nor SUR-6 was required for meiotic spindle assembly. It is likely that the role of PP2A on spindle assembly is achieved by a combination of B subunits and that could also include uncharacterised B subunits (*UniProt Consortium et al., 2020*).

## B56 targeting during different stages of meiosis I

Chromosomal PPTR-1 and PPTR-2 and the ensuing PP2A targeting prior to chromosome segregation depend on the BUB-1 LxxIxE motif. Interestingly, this dependency changes during anaphase, where PPTR-2 chromosomal targeting is BUB-1 dependent but LxxIxE motif independent and depends on the kinase domain. The regulation of protein localisation during meiosis I in *C. elegans* oocytes is highly dynamic, and differences have been observed between metaphase and anaphase. For example, while kinetochore localisation of the CLASP orthologue CLS-2 during metaphase depends on BUB-1, a pool of CLS-2 is able to localise in the anaphase central spindle in a BUB-1-independent manner (*Laband et al., 2017*). In this line, since its role during meiosis was first described (*Dumont et al., 2010*), there has been no clear explanation as to the cause of the alignment and segregation defects upon BUB-1 depletion. Our results provide a plausible explanation for the role of BUB-1 in chromosome alignment during oocyte meiosis in *C. elegans* through the recruitment of PP2A:B56.

## Possible function of chromosome-associated PP2A

As mentioned above, the PP2A complexes are likely playing a number of important roles during meiosis. Our work uncovers a function during chromosome congression involving B56 subunits targeted by BUB-1. Similar to mice and yeast, it is therefore likely that PP2A complexes at the chromosome control proper chromosome-spindle attachments. The alignment defects we observed upon PPTR-1/2 depletion (or in the LxxIxE motif mutant) resemble those reported in the absence of kinetochore proteins (*Danlasky et al., 2020*; *Dumont et al., 2010*). Interestingly, PP2A:B56 complexes containing PPTR-1 and PPTR-2 are present at the meiotic kinetochore, and it is therefore possible that PP2A:B56 works in parallel to and/or regulates kinetochore protein(s). Furthermore, we detected GFP::PAA-1 on anaphase chromosomes, similar to kinetochore proteins (*Danlasky et al., 2020*; *Dumont et al., 2010*; *Hattersley et al., 2016*).

During meiosis in *C. elegans*, the Aurora B orthologue, AIR-2, concentrates in the interface between homologous chromosomes (i.e. the midbivalent; *Kaitna et al., 2002*; *Rogers et al., 2002*; *Schumacher et al., 1998*), and its activity is counteracted by PP1, which is recruited by the protein LAB-1 (*Tzur et al., 2012*). PP1 counteracts Aurora B$^{AIR-2}$ during meiosis by antagonising Haspin-mediated H3T3 phosphorylation in the long arm of the bivalent (*Ferrandiz et al., 2018*). Consistent with this, PP1 depletion leads to loss of sister chromatid cohesion during meiosis I (*Kaitna et al., 2002*; *Rogers et al., 2002*). During prometaphase and metaphase I, Aurora B$^{AIR-2}$ is retained in the bivalent, whereas PP1 resides mainly in kinetochores (*Hattersley et al., 2016*) and it is therefore likely that other phosphatase(s) will be involved in controlling Aurora B-mediated phosphorylation events during metaphase I. Since PP2A:B56 is concentrated in the midbivalent during meiosis I, we hypothesise that it could be balancing AIR-2 activity during meiosis. In this respect, we found that serine 612 in BUB-1 is phosphorylated and this serine is embedded in a sequence (RRL<u>S</u>I) closely resembling the consensus for PP2A:B56 reported in *Kruse et al., 2020*. Furthermore, the sequence also fits into the loosely defined Aurora B consensus RRxSφ (where φ is a hydrophobic aa) (*Cheeseman et al., 2002*; *Deretic et al., 2019*; *Kettenbach et al., 2011*). Therefore, BUB-1 itself could be subjected to an Aurora B-PP2A:B56 balance and it will be very interesting to address whether BUB-1 and other meiotic proteins are regulated by the balance between Aurora B and PP2A:B56.

In a broader context, a putative LxxIxE motif has been identified in human Bub1 (*Cordeiro et al., 2020*; *Singh et al., 2020*; *Wang et al., 2016*). While this motif has not been tested functionally, it is interesting that this putative LxxIxE motif is preceded by a PLK1 binding motif and is important for restraining PLK1 activity during SAC activation (*Cordeiro et al., 2020*). While we did not find a PLK1 binding site preceding BUB-1 LxxIxE, a combination of PLK1 and B56 motifs is present in aa 516–536 of *C. elegans* BUB-1 (*Cordeiro et al., 2020*). It will be interesting to determine whether BUB-1 plays a role in regulating the PLK1-PP2A:B56 balance and also what biological role this axis plays during meiosis.

In summary, we provide evidence for a novel, BUB-1-regulated role for PP2A:B56 during female meiosis in *C. elegans.* It will be interesting to test in the future our hypothesis that PP2A is the main phosphatase counteracting Aurora B-mediated phosphorylation to achieve proper phosphorylation levels during meiosis I.

## Materials and methods

### *C. elegans* strains

Strains used in this study were maintained at 20 degrees unless indicated otherwise. For a complete list of strains, please refer to *Supplementary file 2*. Requests for strains not deposited in the CGC should be done through the FP lab's website (https://pelischlab.co.uk/reagents/).

### RNAi

For RNAi experiments, we cloned the different sequences in the L4440 RNAi feeding vector (*Timmons and Fire, 1998*).

All sequences were inserted into L4440 using the NEBuilder HiFi DNA Assembly Master Mix (New England Biolabs) and transformed into DH5a bacteria. The purified plasmids were then transformed into HT115(DE3) bacteria (*Timmons et al., 2001*). RNAi clones were picked and grown overnight at 37°C in LB (Luria–Bertani medium) with 100 µg/ml ampicillin. Saturated cultures were diluted 1:100 and allowed to grow until reaching an $OD_{600}$ of 0.6–0.8. Isopropyl-β-d-thiogalactopyranoside (IPTG) was added to a final concentration of 1 mM, and cultures were incubated overnight at 20°C. Bacteria were then seeded onto NGM plates made with agarose and allowed to dry. L4 worms were then plated on RNAi plates and grown to adulthood at 20°C for 24 hr in the case of *let-92(RNAi)*, *bub-1(RNAi)*, and *paa-1(RNAi)* and 48 hr in all other cases.

### CRISPR strains

#### GFP::SUR-6

For the generation of in situ-tagged GFP::SUR-6, we used the self-excising cassette method (*Dickinson et al., 2015*). In brief, N2 adults were injected with a plasmid mix containing Cas9, a sgRNA targeting the N-terminus of sur-6 and a repairing template to insert the GFP sequence, along with a LoxP-flanked cassette that encoded for a hygromycin resistance gene, a sqt-1 mutant to confer a dominant roller phenotype, and heat-shock-induced Cre recombinase. After selection in hygromycin, positive integrants (evidenced by their roller phenotype) were heat-shocked to express Cre and remove the cassette.

The strains AID::GFP::GSP-2, GFP::PAA-1, BUB-1[K718R,D847N], BUB-1[L282A,V285A], BUB-1[S283A], and BUB-1[L282A,V285A, K718R,D847N] were generated by Sunybiotech.

#### AID::GFP::GSP-2

(AID in purple, GFP in green, synonimous mutations in cyan).

ATGcctaaagatccagccaaacctccggccaaggcacaagttgtgggatggccaccggtgagatcataccggaagaacgtgatggtttcctgccaaaaatcaagcggtggcccggaggcggcggcgttcgtgaagAGTAAAGGAGAAGAACTTTTCACTGGAGTTGTCCCAATTCTTGTTGAATTAGATGGTGATGTTAATGGGCACAAATTTTCTGTCAGTGGAGAGGGTGAAGGTGATGCAACATACGGAAAACTTACCCTTAAATTTATTTGCACTACTGGAAAACTACCTGTTCCATGGgtaagtttaaacatatatatactaactaaccctgattatttaaattttcagCCAACACTTGTCACTACTTTCTgTTATGGTGTTCAATGCTTcTCgAGATACCCAGATCATATGAAACgGCATGACTTTTTCAAGAGTGCCATGCCCGAAGGTTATGTACAGGAAAGAACTATATTTTTCAAAGATGACGGGAACTACAAGACACgtaagtttaaacagttcggtactaactaaccatacatatttaaattttcagGTGCTGAAGTCAAGTTTGAAGGTGATACCCTTGTTAATAGAATCGAGTTAAAAGGTATTGATTTTAAAGAAGATGGAAACATTCTTGGACACAAATTGGAATACAACTATAACTCACACAATGTATACATCATGGCAGACAAACAAAAGAATGGAATCAAAGTTgtaagtttaaacatgattttactaactaactaatctgatttaaattttcagAACTTCAAAATTAGACACAACATTGAAGATGGAAGCGTTCAACTAGCAGACCATTATCAACAAAATACTCCAATTGGCGATGGCCCTGTCCTTTTACCAGACAACCATTACCTGTCCACACAATCTGCCCTTTCGAAAGATCCCAACGAAAAGAGAGACCACATGGTCCTTCTTGAGTTTGTAACAGCTGCTGGGATTACACATGGCATGGATGAACTATACAAAGACGTAGAAAAGCTTAAT<u>CTC</u>GACAATATCATCTCCAGATTATTGGAAG.

#### GFP::PAA-1

(GFP in green, synonimous mutations in cyan).

ATGAGTAAAGGAGAAGAACTTTTCACTGGAGTTGTCCCAATTCTTGTTGAATTAGATGGTGATGTTAATGGGCACAAATTTTCTGTCAGTGGAGAGGGTGAAGGTGATGCAACATACGGAAAAC

TTACCCTTAAATTTATTTGCACTACTGGAAAACTACCTGTTCCATGGgtaagtttaaacatatatatactaac-
taaccctgattatttaaattttcagCCAACACTTGTCACTACTTTCTgTTATGGTGTTCAATGCTTcTCgAGA
TACCCAGATCATATGAAACgGCATGACTTTTTCAAGAGTGCCATGCCCGAAGGTTATGTACAG-
GAAAGAACTATATTTTTCAAAGATGACGGGAACTACAAGACACgtaagtttaaacagttcggtactaactaac-
catacatatttaaattttcagGTGCTGAAGTCAAGTTTGAAGGTGATACCCTTGTTAATAGAATCGAG
TTAAAAGGTATTGATTTTAAAGAAGATGGAAACATTCTTGGACACAAATTGGAATACAACTATAAC
TCACACAATGTATACATCATGGCAGACAAACAAAGAATGGAATCAAAGTTgtaagtttaaacatgattt-
tactaactaactaatctgatttaaattttcagAACTTCAAAATTAGACACAACATTGAAGATGGAAGCGTTCAAC
TAGCAGACCATTATCAACAAAATACTCCAATTGGCGATGGCCCTGTCCTTTTACCAGACAACCA
TTACCTGTCCACACAATCTGCCCTTTCGAAAGATCCCAACGAAAAGAGAGACCACATGGTCCTTC
TTGAGTTTGTAACAGCTGCTGGGATTACACATGGCATGGATGAACTATACAAAGGAGGTGGA
TCCGGTGGTGGATCCTCGGTTGTCGAAGAAGCTACTGACGACGCG.

## bub-1<sup>K718R,D847N</sup>

The wild-type sequence:
GTAACCGATGATCAAAGGACAGTAGCTGTGAAGTACGAGGTGCCATCATGTTCG
TGGGAAGTGTACATTTGCGACCAAATGCGGAATCGCCTGAAAGATCGAGGTTTGGAGC
TGATGGCCAAATGTTGCATTATGGAAGTGATGGATGCTTATGTTTATTCAACTGCTTCGCTTC
TTGTTAATCAGTACCACGAATATGGAACGCTGCTTGAATATGCGAATAACATGAAGGA
TCCGAATTGGCACATAACCTGCTTCTTGATTACCCAAATGGCCCGAGTTGTGAAGGAAG
TCCATGCCTCTAAAATTATTCATGGAGATATCAAACCGGATAATTTTATGATCACCAGAAA
gtatgggaaaacatttgttaattttagacgttatcttttttcagGATCGATGATAAATGGGGCAAAGATGCTCTGA
TGAGTAACGACAGCTTTGTCATCAAGATTATCGACTGGGGACGTGCCATTGACATGA
TGCCACTGAAGAACCAGCGT
was mutated to:
GTAACCGATGATCAAAGGACAGTAGCTGTGCGCTACGAGGTGCCATCATGTTCG
TGGGAAGTGTACATTTGCGACCAAATGCGGAATCGCCTGAAAGATCGAGGTTTGGAGC
TGATGGCCAAATGTTGCATTATGGAAGTGATGGATGCTTATGTTTATTCAACTGCTTCGCTTC
TTGTTAATCAGTACCACGAATATGGAACGCTGCTTGAATATGCGAATAACATGAAGGA
TCCGAATTGGCACATAACCTGCTTCTTGATTACCCAAATGGCCCGAGTTGTGAAGGAAG
TCCATGCCTCTAAAATTATTCATGGAGATATCAAACCGGATAATTTTATGATCACCAGAAA
gtatgggaaaacatttgttaattttagacgttatcttttttcagGATCGATGATAAATGGGGCAAAGATGCTCTGA
TGAGTAACGACAGCTTTGTCATCAAGATTATCAATGGGGACGTGCGATTGACATGA
TGCCACTGAAGAACCAGCGT

The introduced changes are shown in red, and synonymous mutations are shown in cyan.

## bub-1<sup>L282A,V285A</sup>

The wild-type sequence:
AACGCCAATCTAAATCCTAGAAGACGTCATCTTTCACCAGTCAGTGAGAAAACGGTTGA
TGATGAGGAGGAAAAG
was mutated to:
AACGCCAATCTAAATCCTAGAAGACGTCATGCATCACCAGCTAGCGAGAAAACGGTTGA
TGATGAGGAGGAAAAG

## bub-1<sup>S283A</sup>

The wild-type sequence:
TTCAACGCCAATCTAAATCCTAGAAGACGT**CAT**CTT**TCA**CCAGTCAGTGAGAAAACGGTT
GATGATGAG
was mutated to:
TTCAACGCCAATCTAAATCCTAGAAGACGT**CAC**CTT**GCG**CCAGTCAGTGAGAAAACGG
TTGATGATGAG

The introduced changes are shown in red, and synonymous mutations are shown in cyan.

See *Supplementary file 3* for the primer sequences used for genotyping.

bub-1<sup>L282A,V285A, K718R,D847N</sup>

The wild-type sequence:
AACGCCAATCTAAATCCTAGAAGACGTCAT**CTT**TCACCA**GTCAGT**GAGAAAACGGTTGA
TGATGAGGAGGAAAAGCGAAGCCGGATTTATTCGCCGCTGGTTGCAACGAAGGATGCTCA-
CAGACCTGCACTTCGGAGCAAAATTGAGAATCCTCCAGCGACAGTGACACTTTCGTCGGA
TACAAAGTCTGCTTCGGAGAAAGATGTTAGTGATTCCGATGATGCAGATGATGATGAAA-
GACTCAAGATTATGACTGCCGGCAGAAAGATGGTAACCCTCCAGACCGTTCCACAAGCA
TATCTTCCAACTATTCAACTGCTTCTGCAAGAACATCAAAGAGTGGAGCTGGATTGGA
TTTGATGGCGGAAAATAAGTGTTTGGAGGCACATGCTATGTTTTCCGACACTGTACATC
TTGCTAGCGAAAAGACAATGGTCCTTGGCGATGATTCTGTCTTCGTTCCAGAAAGATC
TTTAGCTACTACGCAGATAGTTACTGACTTTTCCGTGCTCTGTGATCCTGATCCGACAA
TGACCATTACACAGGAGCGTCCGAAAAAAGTGTCGAATGGGTTGAATGTTGTTTACGA
TGAGGCAGCCGAACCGGAAGAATCTCAGAAAGTTGAGGAATCTGAAGTACAACCCGAAA
TTGTCCTAGTTTCTCCAGTGACGCAAACCTCACCAGCTACAATGTTTAATGATAGTGGGTTTA
TCGAAAAATATAACAACTTATGTTTTAATTTTTAGTTTATGACGATGAAATCGAGTTTGGC
TTTTTCAAACCGTCTCGTGGTAATTTCGTCACATCGACCCCCGCACAAGGAGTTCATTTGG
TCAACATTGATGAATATTTCGGAAATAAAGAGGAGGAAAGCACTCACGAACAGGAAGC
TCCAGTATTTGTTGCTCCAACCAGCAGTACTTTCAGTAAATTAGTAAGTGCCAGACAAA
TTTTCGACATACTATTCAAACTTTTTCAGACACGTCGAAAGTCACTAGCAGCAAA
TCAAGCCGTTCAGCCCTCAGTCACAGAGTCATCAAAGCCTGAACGATCAGATCCTAAAGA
TTCATCTATCGATTGTTTGACAGCTAATCTAGGAAGACGTCTTTCAATTGGTGCTGATGAAA
TTCCAAATCTCACTGAAAACAACGAATCTGAAATCACTGGTTGCAAGATTCGTCGGCGCAG
TGAAATTATCAAGCAAGGAGACATCAATCCATGGGACGAAACTCTTCGAAAAAAATTGATG
TGTCTTGTGCGTCCTCCCCAGAATATGCACGAGTTCCAAGAACGAGCACCGAAGA
TTCAAGCTCTGAGAGACTGCGAGGTTAGCGGAGAAAAGCTCCACATTCAAACTCTTATTGG
TCAAGGTGGATACGCTAAAGTATACCGGGCTGTAACCGATGATCAA**AGG**ACAGTAGCTGT-
G**AAG**TACGAGGTGCCATCATGTTCGTGGGAAGTGTACATTTGCGACCAAATGCGGAA
TCGCCTGAAAGATCGAGGTTTGGAGCTGATGGCCAAATGTTGCATTATGGAAGTGATGGA
TGCTTATGTTTATTCAACTGCTTCGCTTCTTGTTAATCAGTACCACGAATATGGAACGCTGC
TTGAATATGCGAATAACATGAAGGATCCGAATTGGCACATAACCTGCTTCTTGATTACC-
CAAATGGCCCGAGTTGTGAAGGAAGTCCATGCCTCTAAAATTATTCATGGAGATA
TCAAACCGGATAATTTTATGATCACCAGAAAGTATGGGAAAACATTTGTTAATTTTAGACG
TTATCTTTTTTCAGGATCGATGATAAATGGGGCAAAGATGCTCTGATGAGTAACGACAGC
TTTGTCATCAAGATTATC**GAC**TGGGGACGT**GCC**ATTGACATGATGCCACTGAAGAAC-
CAGCGT
was mutated to:
AACGCCAATCTAAATCCTAGAAGACGTCAT**GCA**TCACCA**GCTAGC**GAGAAAACGGTTGA
TGATGAGGAGGAAAAGCGAAGCCGGATTTATTCGCCGCTGGTTGCAACGAAGGATGCTCA-
CAGACCTGCACTTCGGAGCAAAATTGAGAATCCTCCAGCGACAGTGACACTTTCGTCGGA
TACAAAGTCTGCTTCGGAGAAAGATGTTAGTGATTCCGATGATGCAGATGATGATGAAA-
GACTCAAGATTATGACTGCCGGCAGAAAGATGGTAACCCTCCAGACCGTTCCACAAGCA
TATCTTCCAACTATTCAACTGCTTCTGCAAGAACATCAAAGAGTGGAGCTGGATTGGA
TTTGATGGCGGAAAATAAGTGTTTGGAGGCACATGCTATGTTTTCCGACACTGTACATC
TTGCTAGCGAAAAGACAATGGTCCTTGGCGATGATTCTGTCTTCGTTCCAGAAAGATC
TTTAGCTACTACGCAGATAGTTACTGACTTTTCCGTGCTCTGTGATCCTGATCCGACAA
TGACCATTACACAGGAGCGTCCGAAAAAAGTGTCGAATGGGTTGAATGTTGTTTACGA
TGAGGCAGCCGAACCGGAAGAATCTCAGAAAGTTGAGGAATCTGAAGTACAACCCGAAA
TTGTCCTAGTTTCTCCAGTGACGCAAACCTCACCAGCTACAATGTTTAATGATAGTGGGTTTA
TCGAAAAATATAACAACTTATGTTTTAATTTTTAGTTTATGACGATGAAATCGAGTTTGGC
TTTTTCAAACCGTCTCGTGGTAATTTCGTCACATCGACCCCCGCACAAGGAGTTCATTTGG
TCAACATTGATGAATATTTCGGAAATAAAGAGGAGGAAAGCACTCACGAACAGGAAGC
TCCAGTATTTGTTGCTCCAACCAGCAGTACTTTCAGTAAATTAGTAAGTGCCAGACAAA
TTTTCGACATACTATTCAAACTTTTTCAGACACGTCGAAAGTCACTAGCAGCAAA
TCAAGCCGTTCAGCCCTCAGTCACAGAGTCATCAAAGCCTGAACGATCAGATCCTAAAGA
TTCATCTATCGATTGTTTGACAGCTAATCTAGGAAGACGTCTTTCAATTGGTGCTGATGAAA
TTCCAAATCTCACTGAAAACAACGAATCTGAAATCACTGGTTGCAAGATTCGTCGGCGCAG
TGAAATTATCAAGCAAGGAGACATCAATCCATGGGACGAAACTCTTCGAAAAAAATTGATG
TGTCTTGTGCGTCCTCCCCAGAATATGCACGAGTTCCAAGAACGAGCACCGAAGA
TTCAAGCTCTGAGAGACTGCGAGGTTAGCGGAGAAAAGCTCCACATTCAAACTCTTATTGG
TCAAGGTGGATACGCTAAAGTATACCGGGCTGTAACCGATGATCAA

**AGA**ACAGTAGCTGTG**CGC**TACGAGGTGCCATCATGTTCGTGGGAAGTGTACATTTGCGAC-
CAAATGCGGAATCGCCTGAAAGATCGAGGTTTGGAGCTGATGGCCAAATGTTGCATTA
TGGAAGTGATGGATGCTTATGTTTATTCAACTGCTTCGCTTCTTGTTAATCAGTACCACGAA
TATGGAACGCTGCTTGAATATGCGAATAACATGAAGGATCCGAATTGGCACATAACCTGC
TTCTTGATTACCCAAATGGCCCGAGTTGTGAAGGAAGTCCATGCCTCTAAAATTATTCA
TGGAGATATCAAACCGGATAATTTTATGATCACCAGAAAGTATGGGAAAACATTTGTTAA
TTTTAGACGTTATCTTTTTTCAGGATCGATGATAAATGGGGCAAAGATGCTCTGATGAG
TAACGACAGCTTTGTCATCAAGATTATC**AAT**TGGGGACGT**GCG**ATTGACATGATGCCAC
TGAAGAACCAGCGT

The introduced changes are shown in red, and synonymous mutations are shown in cyan.

See *Supplementary file 3* for the primer sequences used for genotyping. Data on brood size and embryo viability is available in *Figure 7—figure supplement 2*.

## Live imaging of oocytes

A detailed protocol for live imaging of *C. elegans* oocytes was used with minor modifications (*Laband et al., 2018*). Fertilised oocytes were dissected and mounted in 5 µl of L-15 blastomere culture medium (0.5 mg/ml inulin; 25 mM HEPES, pH 7.5 in 60% Leibowitz L-15 medium and 20% heat-inactivated fetal bovine serum) on 24 × 40 mm #1.5 coverslips. Once dissection was performed and early oocytes identified using a stereomicroscope, a circle of Vaseline was laid around the sample, and a custom-made 24 × 40 mm plastic holder (with a centred window) was placed on top. The sample was imaged immediately. Live imaging was done using a 60×/NA 1.4 oil objective on a spinning-disk confocal microscope (MAG Biosystems) mounted on a microscope (IX81; Olympus), an EMCCD Cascade II camera (Photometrics), spinning-disk head (CSU-X1; Yokogawa Electric Corporation). Acquisition parameters were controlled by MetaMorph seven software (Molecular Devices). For all live imaging experiments, partial maximum-intensity projections are presented in the figures and full maximum-intensity projections are presented in the supplementary videos. All files were stored, classified, and managed using OMERO (*Allan et al., 2012*). Figures were prepared using OMERO.figure and assembled using Adobe Illustrator. Representative videos shown in Supplementary material were assembled using custom-made macros in Fiji/ImageJ (*Schindelin et al., 2012*).

## Generation of phospho-Ser 283 BUB-1 antibody

The antibody was generated by Moravian Biotec by immunising rabbits with the following peptide: RRRHL(pS)PVSEKTC. Serum was adsorbed with a non-phosphorylated peptide (RRRHLSPVSEKTC) followed by affinity purification with the antigenic, phosphorylated peptide. Different fractions were tested in immunofluorescence by incubating different dilutions (1:1,000 and 1:10,000) with 1 µM and 10 µM of either non-phosphorylated or phosphorylated peptide. Only the phosphorylated peptide competed out the signal. Additionally, we used the BUB-1$^{S283A}$ strain, and no antibody signal was detected.

## Immunofluorescence

Worms were placed on 4 µl of M9 worm buffer in a poly-D-lysine (Sigma, P1024)-coated slide, and a 24 × 24 cm coverslip was gently laid on top. Once the worms extruded the embryos, slides were placed on a metal block on dry ice for >10 min. The coverslip was then flicked off with a scalpel blade, and the samples were fixed in methanol at 20°C for 30 min. After blocking in phosphate-buffered saline (PBS) buffer plus 3% bovine serum albumin and 0.1% Triton X-100 (AbDil), samples were incubated overnight at 4°C with anti-BUB-1 (*Desai et al., 2003*) and anti-tubulin (1/400, clone DM1α, Sigma–Aldrich) in AbDil. After three washes with PBS plus 0.1% Tween, secondary antibodies were added at 1/1000 (goat anti-mouse and goat anti-rabbit conjugated to Alexa Fluor 488, Alexa Fluor 594, Thermo Scientific). After 2 hr at room temperature and three washes with PBS plus 0.1% Tween, embryos were mounted in ProLong Diamond antifade mountant with DAPI (Thermo Scientific).

For the comparison of total versus phospho Ser 283 BUB-1, a total BUB-1 antibody was labelled with Alexa-488, while phospho Ser 283 BUB-1 was labelled with Alexa 647, using the APEX Alexa Fluor Antibody Labelling kits (Thermo). We used the strain HY604, which is a temperature-sensitive allele of the the APC component MAT-1, that arrests in meiosis I prior to spindle rotation when moved to the restrictive temperature.

## GFP immunoprecipitation

For GFP immunoprecipitations, we followed a published protocol (*Sonneville et al., 2017*) with minor modifications (*Pelisch et al., 2019*). Approximately 1000 worms expressing GFP-tagged endogenous BUB-1 were grown for two generations at 20°C in large 15 cm NGM plates with concentrated HT115 bacteria. Worms were bleached and embryos were laid in new 15 cm NGM plates with concentrated HT115 bacteria. Once >80% of the worm population was at the L3/L4 stage, worms were washed and placed on 15 cm agarose plates containing concentrated HT115 bacteria. After 24 hr, worms were bleached and the embryos were resuspended in a lysis buffer containing 100 mM HEPES–KOH pH 7.9, 50 mM potassium acetate, 10 mM magnesium acetate, 2 mM ethylenediamine-tetraacetic acid (EDTA), 1× protease inhibitor ULTRA (Roche), 2× PhosSTOP (Roche), and 1 mM dithiothreitol (DTT). The solution was added drop-wise to liquid nitrogen to generate beads that were later grinded using a SPEX SamplePrep 6780 Freezer/Mill. After thawing, we added one-quarter volume of buffer containing lysis buffer supplemented with 50% glycerol, 300 mM potassium acetate, 0.5% NP40, plus DTT, protease, and phosphatase inhibitors as described above. DNA was digested with 1600U of Pierce Universal Nuclease for 30 min on ice. Extracts were centrifuged at 25,000 g for 30 min and then at 100,000 g for 1 hr. The extract was then incubated for 60 min with 30 µl of a GFP nanobody covalently coupled to magnetic beads. The beads were washed 10 times with 1 ml of wash buffer (100 mM HEPES–KOH pH 7.9, 300 mM potassium acetate, 10 mM magnesium acetate, 2 mM EDTA, 0.1% NP40, plus protease and phosphatase inhibitors) at 4°C (cold room). Bound proteins were eluted twice using two rounds of 50 µl LDS (Lithium dodecyl sulfate) sample buffer (Thermo Scientific) at 70°C for 15 min and stored at −80°C.

## Sample preparation for mass spectrometry

IP samples were run on 4–12% Bis–Tris sodium dodecyl sulfate gels with MOPS running buffer, and the gel was stained using Quick Coomassie Stain (Generon). Bands of interest were cut and washed with water:acetonitrile (50:50), followed by a wash with 100 mM ammonium bicarbonate. The gel pieces were then washed with 100 mM ammonium bicarbonate:acetonitrile (50:50), followed by a final wash with acetonitrile. Gel pieces were dried using a SpeedVac.

Samples were reduced with 10 mM DTT in 20 mM ammonium bicarbonate and alkylated with 50 mM IAA (iodoacetamide) in 20 mM ammonium bicarbonate. Samples were then washed sequentially with 100 mM ammonium bicarbonate, 100 mM ammonium bicarbonate:acetonitrile (50:50), and acetonitrile. Gel pieces were dried using a SpeedVac.

Trypsin solution (12.5 µg/ml stock in 20 mM ammonium bicarbonate) was added to cover the gel pieces and incubated for 30 min on a shaking platform and incubated sample overnight at 30°C on a shaker. Peptides were extracted by standard procedures and reconstituted in 10 µl of 5% formic acid/10% acetonitrile. After vortexing for 1 min, water was added to 50 µl.

## Mass spectrometry analysis

Samples were run on an Ultimate 3000 RSLCnano system (ThermoFisher Scientific) coupled with a Q-Exactive Plus Mass Spectrometer (Thermo Fisher Scientific). Peptides initially trapped on an Acclaim PepMap 100 (Thermo Fisher Scientific) and then separated on an Easy-Spray PepMap RSLC C18 column (Thermo Fisher Scientific). Sample was transferred to mass spectrometer via an Easy-Spray source with temperature set at 50°C and a source voltage of 2.1 kV. The mass spectrometer was operated on in data-dependent acquisition mode (top 15 method). MS resolution was 70,000 with a mass range of 350–1600. MS/MS resolution was 17,500.

RAW data files were extracted and converted to mascot generic files (.mgf) using MSC Convert. Extracted data then searched against *C. elegans* proteome and BUB-1 specifically, using the Mascot Search Engine (Mascot Daemon Version 2.3.2). Type of search used was MS/MS Ion Search using Trypsin/P. Carbamidomethyl (C) was set as a fixed modification, and variable modifications were as follows: acetyl (N-term), dioxidation (M), Gln to pyro-Glu (N-term Q), oxidation (M), deamidation (NQ), and phosphorylation (STY).

Peptide mass tolerance was ±10 ppm (# 13C = 2) with a fragment mass tolerance of ±0.6 Da. Maximum number of missed cleavages was 2.

The mass spectrometry proteomics data have been deposited to the ProteomeXchange Consortium via the PRIDE (*Perez-Riverol et al., 2019*) partner repository with the dataset identifier PXD023258.

## Sequence alignment

All sequence alignments were performed using Clustal Omega (*Sievers et al., 2011*), version 1.2.4. Full-length mammalian B56α (UniProtKB Q15172), β (UniProtKB Q15173), γ (UniProtKB Q13362), δ (UniProtKB Q14738), and ε (UniProtKB Q16537) alongside their alternative splicing isoforms were retrieved from UniProt (*The UniProt Consortium, 2017*) and aligned with full-length *C. elegans* PPTR-1 (UniProtKB O18178) and PPTR-2 (UniProtKB A9UJN4-1). A guide tree was calculated from the distance matrix generated from sequence pairwise scores.

The C-terminal regions of mammalian isoforms B56γ (S378-A393), δ (S454-A469), β (S409-V424), α (S403-V418), ε (S395-V410) and *C. elegans* PPTR-1 (S420-V435) and PPTR-2 (S449-A464) were used for the alignment. The canonical sequences of each B56 isoform were retrieved from UniProt (*UniProt Consortium et al., 2020*).

The SLiMs of *C. elegans* BUB-1 (R279-D292;UniProtKB Q21776), human BubR1 (I666-A679;UniProtKB O60566-1), and RepoMan (K587-P600; UniProtKB Q69YH5-1) were aligned with Clustal Omega (*Sievers et al., 2011*) and visualised with Jalview (*Waterhouse et al., 2009*).

## Kinase assays

Forty microlitre reactions were set up containing 40 mM Tris–HCl pH 7.5, 100 µM ATP (Adenosine triphosphate), 10 mM $MgCl_2$, and 2 µg of either BUB-1 (259–332) or H1. At t = 0, 15 ng/µl Cdk1/Cyclin B was added, before incubation at 30°C for 30 min. Aliquots were taken immediately after Cdk1/CyclinB addition at t = 0 and then again after the 30 min incubation. All samples were then incubated at 70°C for 15 min in a final concentration of 1× LDS buffer. Sodium dodecyl sulfate–polyacrylamide gel electrophoresis (SDS-PAGE) was then conducted on a NuPage 4–12% Bis–Tris gel (Thermo) with MES buffer before being stained with ProQ Diamond (Thermo) and imaged using Bio-Rad ChemiDoc. Once fluorescence was recorded, Coomassie staining was performed. For the western blot, SDS–PAGE was conducted as above with 67 ng of substrate protein per well before the western was conducted using a nitrocellulose membrane (GE Healthcare) and 1× NuPage transfer buffer (Thermo). The membrane was blocked using Intercept PBS blocking buffer (LI-COR), the primary antibodies used were anti-GST at 1:1000 (made in sheep), and anti-phospho Ser 283 1:20,000 (made in rabbit). Secondary antibodies were anti-sheep IRDye 680RD anti-rabbit 800CW (LI-COR), both at 1:50,000. The membrane was then imaged using LI-COR Odyssey CLx.

## Expression and purification of PPTR-2

PPTR-2 was expressed from pHISTEV30a vector as a 6xHis- and Strep-tagged protein (plasmid fgp_445) in *Escherichia coli* strain BL21 (DE3) and the bacterial culture was incubated overnight at 37°C with shaking at 220 rpm. Bacteria were grown in TB medium at 37°C with shaking at 220 rpm until OD600 reached ~0.6–0.8 and induced with 150 µM IPTG. Induction was performed at 18°C with shaking at 220 rpm for ~16 hr. Cells were pelleted and lysed by sonication in 50 mM NaP, 300 mM NaCl, 10 mM imidazole, 10% glycerol, 0.5 mM TCEP, (tris(2-carboxyethyl)phosphine) and protease inhibitors. After binding to a Ni-NTA column, protein was washed with 20 mM imidazole and then eluted with 350 mM imidazole. The tag was cleaved with 6xHis-tagged TEV (Tobacco Etch Virus Protease) protease overnight at 4°C, and the tag and protease were removed from the sample by binding to Ni-NTA. PPTR-2 was concentrated and further purified using a HiLoad 16/600 Superdex 200 pg size exclusion column.

## Fluorescence polarisation

The following peptides were synthesised by peptides and elephants GmbH: BUB-1 FITC-Ahx-NPRRRHLSPVSEKTVDDEEE, pBUB-1 FITC-Ahx-NPRRRHLphSPVSEKTVDDEEE, and LAVA FITC-Ahx-NPRRRHASPASEKTVDDEEE. Reactions (35 µl) were set up in FP buffer (50 mM NaP pH 7.5, 150 mM NaCl, 0.5 mM TCEP) containing peptide concentrations 0.5–1 µM. These were then used to create a 1:2 serial dilution series using FP buffer containing the same peptide concentrations as above as well as the indicated concentration of PPTR-2. Reactions were left for 30 min before triplicates of each

dilution were aliquoted into a black 384-well plate, centrifuged (2 min, 2000 rpm), and analysed using the PheraStar FS.

## Image analysis and statistics

For time-dependent analysis, metaphase I was taken as time = 0 s. This frame was chosen as the previous frame where the first indication of chromosome separation was visible. All image analysis was done in Fiji (*Schindelin et al., 2012*). For total intensity measurements within the meiotic spindle, images were thresholded and binary images were generated using the ReinyEntropy method and used to automatically generate the regions of interest (ROIs) for each slice (z) and time point (t) or in the sum-projected image. Going through all the slices was particularly necessary when measuring protein intensity associated with a specific structure/location, since the angle of the spindle can lead to erroneous analysis. All ROIs were recorded and, in order to get background measurements, the ROIs were automatically placed on a different oocyte region, away from the spindle. Background-substracted intensity values were then normalised to the maximum intensity value of the control video using Graphpad Prism 7.0 and are presented as mean ± s.e.m.

Central spindle to chromosome ratio for PPTR-2::GFP (*Figure 4G*) was obtained as follows. Images were selected at early anaphase (t = 40 s), and to obtain the chromosome ROIs, we used the mCherry::histone channel to create a mask. These ROIs were transferred to the PPTR-2::GFP channel, and the intensity was measured as described above. The region between the chromosomes was selected intensities were measured to obtain the central spindle intensity. Background corrected values were used to obtain the ratio central spindle/chromosome. Results are shown as median with interquartile range, and differences were analysed using an unpaired two-tailed t-test with Welch's correction.

For the alignment/congression analysis, we selected videos in which the spindles were contained in a single Z-plane at −80 s. There, we established the spindle axes with a line extending from pole to pole and the 'metaphase plate' with a perpendicular line in the middle of the spindle (see *Figure 2—figure supplement 3*). A line was drawn on the long axes of the bivalents, and the angle between this line and the spindle axis was measured ('θ'). Additionally, for each bivalent, the distance ('d') between the centre of the bivalent and the metaphase plate was measured.

Contingency tables were analysed using the Fisher's exact test (two tailed), and the p values are presented in the figures and/or in the text.

## Generation of supplementary videos

The 4D TIFF files were converted to video (.avi) files using a custom-made macro that uses the Stack-Reg Fiji plugin for image registration. Videos were assembled using maximum-intensity projections; hence, the videos might not match a specific panel within the main figures, which are single slices or partial projections in order to highlight specific characteristics.

## Acknowledgements

We would like to thank Peter Askjaer, Guy Benian, Needhi Bhalla, Bruce Bowerman, Arshad Desai, Tony Hyman, and Enrique Martinez-Perez for sharing *C. elegans* strains and antibodies. We would like to thank Egon Ogris for sharing unpublished data on the use of the PP2Ac monoclonal antibody. We thank Arshad Desai, Ron Hay, and Tomo Tanaka for comments on the manuscript. This work was supported by a Career Development Award from the Medical Research Council (grant MR/R008574/1) and an ISSF grant funded by the Wellcome Trust (105606/Z/14/Z). ST is funded by a Medical Research Council Doctoral Training Programme. DKC is supported by a Sir Henry Dale Fellowship from the Wellcome Trust (208833). PL-G and JB were supported by NIH grant R01 GM074215, awarded to Arshad Desai. Work in the JNB lab is supported by NIH grant R01 GM114471. We acknowledge the FingerPrints Proteomics Facility and the Dundee Imaging Facility, which are supported by a 'Wellcome Trust Technology Platform' award (097945/B/11/Z) and the Tissue Imaging Facility, funded by a Wellcome Trust award (101468/Z/13/Z). Some nematode strains were provided by the CGC, which is funded by NIH Office of Research Infrastructure Programs (P40 OD010440).

## Additional information

### Funding

| Funder | Grant reference number | Author |
|---|---|---|
| Medical Research Council | MR/R008574/1 | Laura Bel Borja<br>Flavie Soubigou<br>Federico Pelisch |
| Wellcome Trust | 208833 | Dhanya K Cheerambathur |
| NIH Office of the Director | R01 GM074215 | Jacqueline Budrewicz<br>Pablo Lara-Gonzalez |
| NIH Office of the Director | R01 GM114471 | Christopher G Sorensen Turpin<br>Joshua N Bembenek |
| Wellcome Trust | 105606/Z/14/Z | Federico Pelisch |
| Medical Research Council | Doctoral Training Programme | Samuel JP Taylor |

The funders had no role in study design, data collection and interpretation, or the decision to submit the work for publication.

### Author contributions

Laura Bel Borja, Flavie Soubigou, Samuel J P Taylor, Formal analysis, Investigation, Methodology, Writing - review and editing; Conchita Fraguas Bringas, Formal analysis, Investigation, Methodology; Jacqueline Budrewicz, Resources, Methodology; Pablo Lara-Gonzalez, Christopher G Sorensen Turpin, Dhanya K Cheerambathur, Resources, Methodology, Writing - review and editing; Joshua N Bembenek, Resources, Investigation, Writing - review and editing; Federico Pelisch, Conceptualization, Data curation, Formal analysis, Supervision, Funding acquisition, Investigation, Methodology, Writing - original draft, Project administration, Writing - review and editing

### Author ORCIDs

Laura Bel Borja (iD) http://orcid.org/0000-0002-8381-934X
Conchita Fraguas Bringas (iD) http://orcid.org/0000-0001-9594-5856
Federico Pelisch (iD) https://orcid.org/0000-0003-4575-1492

### Decision letter and Author response

Decision letter https://doi.org/10.7554/eLife.65307.sa1
Author response https://doi.org/10.7554/eLife.65307.sa2

## Additional files

### Supplementary files

• Supplementary file 1. Percent Identity (%) Matrix of the full length sequence alignment of mammalian B56 isoforms and *C. elegans* orthologues PPTR-1 and PPTR-2. Created with Clustal Omega version 2.1.

• Supplementary file 2. List of *C. elegans* strains used in this study.

• Supplementary file 3. List of primers used for genotyping.

• Transparent reporting form

### Data availability

While some time points are shown in the figures, representative movies showing all time points are provided as Supplementary Movies. The mass spectrometry proteomics data have been deposited to the ProteomeXchange Consortium via the PRIDE (Perez-Riverol et al., 2019) partner repository with the dataset identifier PXD023258.

The following dataset was generated:

| Author(s) | Year | Dataset title | Dataset URL | Database and Identifier |
|---|---|---|---|---|
| Pelisch F, Taylor SJ | 2020 | BUB-1::GFP phosphorylation sites in embryo extracts | https://www.ebi.ac.uk/pride/archive/projects/PXD023258 | PRIDE, PXD023258 |

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
