## [Decision Letter]

**Acceptance summary:**

Using a combination of biochemistry, genetics and live imaging, Borja et al. show that the kinase BUB-1 recruits, through two regulatory subunits, the phosphatase PP2A to meiotic spindle chromosomes during oocyte meiosis I to promote proper congression of the chromosomes prior to anaphase. This recruitment occurs independently of the conserved protein Shugoshin, which has been shown by others to promote PP2A recruitment to chromosomes during mouse oocyte meiosis I. Moreover, Borja et al. show that phosphorylation of a peptide motif in BUB-1 promotes this recruitment, and that this BUB-1motif is likely targeted by CDK-1 for phosphorylation to provide proper temporal regulation of these events.

**Decision letter after peer review:**

[Editors’ note: the authors submitted for reconsideration following the decision after peer review. What follows is the decision letter after the first round of review.]

Thank you for submitting your work entitled "PP2A:B56 Regulates Meiotic Chromosome Segregation in *C. elegans* Oocytes" for consideration by *eLife*. Your article has been reviewed by three peer reviewers, and the evaluation has been overseen by a Reviewing Editor and a Senior Editor. The following individuals involved in review of your submission have agreed to reveal their identity: Jakob Nilsson (Reviewer #2); Patrick Meraldi (Reviewer #3).

Our decision has been reached after consultation between the reviewers. Based on these discussions and the individual reviews below, we regret to inform you that your work will not be considered further for publication in *eLife*.

You will see from the reviews that all the reviewers agree that your study will be of interest to the meiosis community in identifying that BUB1 rather than Sgo recruits PP2A, but that also all agree that more experiments are required to provide firm evidence for a number of your conclusions that would be necessary for publication in *eLife*; in particular whether BUB1 directly binds to PP2A, through the LxxIxE motif, and whether Cdk phosphorylation of this is important. After discussion, they concluded that the amount of time and effort that will be required for these experiments is such that the fairest thing to do is to return the manuscript to you.

Reviewer #1:

Borja et al. describe their analysis of the requirements for the phosphatase PP2A during *C. elegans* oocyte meiotic cell division. They show that PP2A and two conserved B56-type regulatory subunits are localized to spindle poles, chromosomes and the central spindle (with some differences for the B56 subunits), and that all required for spindle assembly, chromosome congression to the metaphase plate, and for chromosome segregation (with some redundancy for the two B56 subunits). The further show that a conserved LxxIxE motif in BUB-1 recruits most of one and some of the other B56 subunit, and that the kinase domain of BUB-1 is required to recruit the other (presumably through an intermediary). Thus in *C. elegans*, a novel form of PP2A recruitment functions, with the other known recruiters, Shugoshin and Mad3 (also with LxxIxE motifs), appearing not be required. The authors use mutational analysis to nicely document the requirements for both parts of BUB-1 in PP2A recruitment, and also identify a phosphorylated residue in BUB-1 that may be involved in CDK-1 regulation of the recruitment. The authors propose that BUB-1 recruitment of PP2A to the central spindle is important for chromosome segregation during anaphase in *C. elegans* oocyte meiosis I.

While the authors provide an extensive analysis of the requirements for PP2A during *C. elegans* oocyte meiosis, with results that will be of substantial interest to investigators studyng oocyte meiotic cell division, the advances are in my opinion to incremental, and also lacking in conclusiveness as to the actual mechanism involved. Therefore as written the manuscript is not suitable for publication in *eLife*, as summarized in the major comments below.

1) The authors provide clear evidence that PP2A is required for spindle assembly, congression of chromosomes to the metaphase plate, and chromosome segregation (with some caveats as to chromosome segregation noted below). While the authors favor and propose that the chromosome segregation defects are due to central spindle defects (with PP2A localizing to the central spindle), they in fact provide no evidence to support this conclusion. It is entirely possible that earlier spindle assembly defects are responsible for the subsequent defects in both congression and segregation. The authors fail to address this possibility and provide no evidence to rule it out. Without some idea as to which targets PP2A acts through, the advance is incremental relative to what is known in other systems, only showing a variation on how LxxIxE motifs can recruit PP2A to spindle structures through BUB-1 instead of other factors. The authors discuss the AuroraB kinase AIR-2 as a possible target but provide no analysis of such a role. Without more mechanistic insight, the manuscript is of substantial interest but seems more appropriate for a more specialized journal such as Molecular Biology of the Cell.

2) In figures throughout the manuscript, the authors use polar body extrusion as a proxy for chromosome segregation defects. However, it is known from work on other *C. elegans* mutants that chromosomes segregation and polar extrusion are not tightly correlated. The authors should establish a more direct approach to quantifying the defects in chromosome segregation; as written only representative examples are shown.

3) The authors show that there are severe spindle assembly defects after knockdown of PP2A but only show microtubules and chromosomes to document the defects. Given that the spindle assembly defects might be primarily responsible for subsequent defects in chromosome alignment and segregation, the authors should better characterize the spindle assembly defects using pole marker(s). Do mutant oocytes even establish a bipolar spindle? From the images, it seems likely they do not.

4) The authors generate point mutations in both the LxxIxE motif and the kinase domain of BUB-1 to nicely document requirements for recruiting PP2A. However, the authors do not include any mention of whether these mutations are essential for oocyte meiotic cell division, or if the mutations are associated with any embryonic lethality (they only show some relatively rare defects in chromosome alignment). In fact, from the supplemental tables it appears that both mutant strains are homozygous viable, although this is never mentioned in the text. If these motifs are indeed responsible for recruiting PP2A and its essential functions to the spindle, then the mutants should result in defects identical to the PP2A knockdowns. Thus to verify the importance of these mutations, the authors would need to construct a balanced strain in which both motifs are mutated and document defects much like those observed in the PP2A knockdowns. More generally, the authors should provide a genetic analysis of the viability of the single mutants (are there reductions in embryonic lethality or brood sizes?).

Reviewer #2:

This manuscript explores female meiosis and the role of Bub1 in targeting PP2A-B56 to chromosomes and central spindle for proper meiosis in *C. elegans*. Although the shugoshin proteins have been shown to be important regulators of cohesin during meiosis by recruiting PP2A-B56 in other systems the authors show this is not the case in *C. elegans*. Instead they show that the Bub1 protein recruits PP2A-B56 through a LxxIxE motif that resembles the one found in human BubR1. Mutation of this motif prevents the recruitment of the PPTR1 (one of the B56 isoforms) to the midbivalent and the central spindle and reduces the recruitment of the PPTR2 to the central spindle. The Bub1 variant with a mutated LxxIxE motif displays misalignments suggesting an imbalance in phosphorylations. The phenotype of the Bub1 mutant is not as severe as Bub1 RNAi but this could be because the kinase domain of Bub1 helps in recruiting PPTR2.

Overall the paper is easy to read and the data are consistent. However, the paper is fairly descriptive and lacks to some degree mechanistic insight. I think that some additional experiments would make the work more interesting for the readers of *eLife*.

1) Characterization of the double Bub1 mutant with a mutated LxxIxE motif and mutations in the kinase domain. This would clarify if there is some redundancy in recruitment mechanisms

2) Direct evidence that the LxxIxE motif of Bub1 facilitates binding to PPTR1/2 – IP or purified components

3) Characterization of the LAPI mutation of the LxxIxE motif to determine if Cdk1 phosphorylation of this site is important. Reading the manuscript as it is now this is purely speculative despite they go through the effort of showing phosphorylation.

Reviewer #3:

The study by Bel Borja and colleagues studies to which extent the PP2A phosphatase contributes to chromosome segregation in *C. elegans* anaphase. It finds that PP2A plays a crucial role in chromosome alignment and segregation. This role depends on the B56 regulatory subunits. Moreover, the author show that the recruitment of the B56 subunits to the spindle apparatus and kinetochores in anaphase depends on the Bub-1 kinase.

The study explores a novel role of PP2A in *C. elegans* anaphase, and is generally of good technical quality (see below), yet at the same time feels a bit thin in terms of results. In particular several major claims of the discussion are not supported by the data. Moreover, it is not clear to which extent the reported experiments are reproducible.

1) The authors do not indicate the number of independent experiments or the number of embryos on which the conclusions are based on. This information is essential to evaluate the reproducibility of the results. The authors use good statistical tests, but the reader must know whether the experiments are based on 5, 10 or 20 embryos.

2) The authors claim the Bub-1 acts on *C. elegans* anaphase by RECRUITING PP2A:B56 to the spindle apparatus. The presented experiments only show that the localization of the B56 depends on Bub-1, which is not the same thing. The authors could test by IP and in vitro pull-down that the claimed interaction is direct (in vitro), or that the proteins are part of the same complex (in vivo). This should be feasible since the authors already have performed IP experiments.

3) The authors claim that PP2A:B56 counteracts Aurora-B during anaphase, but have no evidence for this claim. Does a Aurora-B hypomorph mutant attenuate the B56 loss phenotype? While a negative result would not mean that the model is wrong, a positive result would certainly help to bolster that claim.

---

## [Author Response]

[Editors’ note: the authors resubmitted a revised version of the paper for consideration. What follows is the authors’ response to the first round of review.]

You will see from the reviews that all the reviewers agree that your study will be of interest to the meiosis community in identifying that BUB1 rather than Sgo recruits PP2A, but that also all agree that more experiments are required to provide firm evidence for a number of your conclusions that would be necessary for publication in eLife; in particular whether BUB1 directly binds to PP2A, through the LxxIxE motif, and whether Cdk phosphorylation of this is important. After discussion, they concluded that the amount of time and effort that will be required for these experiments is such that the fairest thing to do is to return the manuscript to you.

The new version of the manuscript has numerous new experiments to address reviewers’ comments/concerns. Relating to the particular two points highlighted by the reviewing editor, we provide data showing that the BUB-1 LxxIxE motif binds the B56 orthologue PPTR-2 in vitro and Ser 283 phosphorylation increases affinity. Furthermore, mutating Ser 283 to Ala drastically inhibits B56 recruitment in vivo resulting in a chromosome alignment defect similar to the L282A,V285A mutant.

We also performed in vitro kinase assays and showed that Cdk1 can phosphorylate Ser 283 in BUB-1 and developed a phospho-specific antibody recognising phosphor-Ser 283 and show that phosphorylated BUB-1 is detected … in vivo.

Thanks to the new quantitative phenotypic analysis we performed, it became clear that B56 subunits play a role during chromosome alignment/congression prior to anaphase. Furthermore, this relies on B56 recruitment by the newly identified BUB-1 LxxIxE motif.

Reviewer #1:[…]1) The authors provide clear evidence that PP2A is required for spindle assembly, congression of chromosomes to the metaphase plate, and chromosome segregation (with some caveats as to chromosome segregation noted below). While the authors favor and propose that the chromosome segregation defects are due to central spindle defects (with PP2A localizing to the central spindle), they in fact provide no evidence to support this conclusion. It is entirely possible that earlier spindle assembly defects are responsible for the subsequent defects in both congression and segregation. The authors fail to address this possibility and provide no evidence to rule it out.

We agree we might have oversimplified the phenotypes and the connection between them. Our biggest mistake was to base the alignment analysis on subjective observations its classification as either “aligned” or “misaligned”.

Firstly, we performed a new analysis related to the chromosome alignment phenotype. This is now detailed in the Materials and methods section and it provides more rigorous and less subjective quantifications which lead us to a better characterisation of this phenotype. This had a tremendous impact on the manuscript, because it is precisely this process of chromosome alignment where BUB-1 targeted PP2A:B56 plays a role. To avoid any potential amplification of the phenotypes due to partial loss of function in PP2A subunits, we decided to measure the phenotypes on the GFP::tubulin expressing strain and not on the strains with tagged PP2A subunits.

PP2A is essential for the assembly of a bipolar spindle (New Figure 1F,G) and chromosome alignment and segregation. While we agree with the reviewer that in this case it is difficult to tease out whether the alignment and segregation are only a direct consequence of this, depletion of the B56 subunits PPTR-1 and PPTR-2 does not affect spindle assembly but perturbs chromosome alignment/congression (New Figure 2E,F). Since this phenotype is measurable and independent from any noticeable spindle defect, we focused on this “clean” phenotype and sought to understand the molecular mechanisms driving it.

Without some idea as to which targets PP2A acts through, the advance is incremental relative to what is known in other systems, only showing a variation on how LxxIxE motifs can recruit PP2A to spindle structures through BUB-1 instead of other factors. The authors discuss the AuroraB kinase AIR-2 as a possible target but provide no analysis of such a role. Without more mechanistic insight, the manuscript is of substantial interest but seems more appropriate for a more specialized journal such as Molecular Biology of the Cell.

While we respect the reviewer’s view on the advance provided by our manuscript, we do believe we provide mechanistic insight into the targeting of PP2A:B56 in a context in which the previously known/characterised regulators (BubR1 and Shugoshin) do not play a role. On the other hand, while BUB-1 is known to be important during female meiosis in *C. elegans*, it was not clear what role(s) BUB-1 play. Therefore, we think we are providing a significant advance on the mechanisms in place to regulate meiosis I. While we are actively working on identifying PP2A substrates and its potential role in antagonising Aurora B, we think this will require an in depth analysis that will be the focus of another paper.

2) In figures throughout the manuscript, the authors use polar body extrusion as a proxy for chromosome segregation defects. However, it is known from work on other *C. elegans* mutants that chromosomes segregation and polar extrusion are not tightly correlated. The authors should establish a more direct approach to quantifying the defects in chromosome segregation; as written only representative examples are shown.

We apologise for the misunderstanding. We did not intend to use PB extrusion as a proxy for chromosome segregation defects, but rather analysed it as an independent process. We are now making this clear in the manuscript. We analyse chromosome alignment defects (see below), lagging chromosomes, and polar body extrusion. For these three phenomena, we provide the quantitative assessment for each condition and show a representative image (or sets of images).

3) The authors show that there are severe spindle assembly defects after knockdown of PP2A but only show microtubules and chromosomes to document the defects. Given that the spindle assembly defects might be primarily responsible for subsequent defects in chromosome alignment and segregation, the authors should better characterize the spindle assembly defects using pole marker(s). Do mutant oocytes even establish a bipolar spindle? From the images, it seems likely they do not.

As requested by the reviewer, we have performed new experiments using GFP-ASPM-1 as a pole marker. As expected, LET-92-depleted oocytes fail to assemble a bipolar spindle (New Figure 1F,G). As mentioned above, in order to differentiate subsequent phenotypes from the spindle defect, we focused on the alignment/congression defects which are observed even in the presence of a seemingly normal bipolar spindle.

4) The authors generate point mutations in both the LxxIxE motif and the kinase domain of BUB-1 to nicely document requirements for recruiting PP2A. However, the authors do not include any mention of whether these mutations are essential for oocyte meiotic cell division, or if the mutations are associated with any embryonic lethality (they only show some relatively rare defects in chromosome alignment). In fact, from the supplemental tables it appears that both mutant strains are homozygous viable, although this is never mentioned in the text. If these motifs are indeed responsible for recruiting PP2A and its essential functions to the spindle, then the mutants should result in defects identical to the PP2A knockdowns. Thus to verify the importance of these mutations, the authors would need to construct a balanced strain in which both motifs are mutated and document defects much like those observed in the PP2A knockdowns. More generally, the authors should provide a genetic analysis of the viability of the single mutants (are there reductions in embryonic lethality or brood sizes?).

Full analysis of embryo viability and brood size analysis is now presented for all the mutants used in the study (Figure 7-figure supplement 2).

Reviewer #2:This manuscript explores female meiosis and the role of Bub1 in targeting PP2A-B56 to chromosomes and central spindle for proper meiosis in *C. elegans*. Although the shugoshin proteins have been shown to be important regulators of cohesin during meiosis by recruiting PP2A-B56 in other systems the authors show this is not the case in *C. elegans*. Instead they show that the Bub1 protein recruits PP2A-B56 through a LxxIxE motif that resembles the one found in human BubR1. Mutation of this motif prevents the recruitment of the PPTR1 (one of the B56 isoforms) to the midbivalent and the central spindle and reduces the recruitment of the PPTR2 to the central spindle. The Bub1 variant with a mutated LxxIxE motif displays misalignments suggesting an imbalance in phosphorylations. The phenotype of the Bub1 mutant is not as severe as Bub1 RNAi but this could be because the kinase domain of Bub1 helps in recruiting PPTR2.Overall the paper is easy to read and the data are consistent. However, the paper is fairly descriptive and lacks to some degree mechanistic insight. I think that some additional experiments would make the work more interesting for the readers of eLife.1) Characterization of the double Bub1 mutant with a mutated LxxIxE motif and mutations in the kinase domain. This would clarify if there is some redundancy in recruitment mechanisms.

We have performed the suggested experiment. The results indicate that LxxIxE motif-mediated recruitment of B56 subunits operates mostly in the midbivalent and central spindle, whereas the kinase domain mediates mostly chromosome chromosomal recruitment of the B56 subunits (New Figure 7).

2) Direct evidence that the LxxIxE motif of Bub1 facilitates binding to PPTR1/2 – IP or purified components.

We have performed in vitro binding experiments and present evidence for the direct interaction between the LxxIxE motif of BUB-1 and B56. We expressed recombinant, full-length PPTR-2 in bacteria and performed fluorescence polarisation experiments using fluorescently labelled LxxIxE motif peptides. We could observe that PPTR-2 bound the wild - type sequence and that L282A,V285A mutations abolished binding (New Figure 4D). On the contrary, binding was enhanced when the peptide was phosphorylated at Serine 283 (New Figure 5F).

3) Characterization of the LAPI mutation of the LxxIxE motif to determine if Cdk1 phosphorylation of this site is important. Reading the manuscript as it is now this is purely speculative despite they go through the effort of showing phosphorylation.

We have now expanded on the LxxIxE motif phosphorylation and its role in vivo.

1) in vitro assays demonstrate that Cdk1 phosphorylates Serine 283 in vitro. This is now part of the New Figure 5C-E.

2) To analyse the role of Serine 283 phosphorylation in vivo, we mutated it to alanine in endogenous BUB-1. PPTR-1 and PPTR-2 localisation to the midbivalent and central spindle was significantly affected in BUB-1^S283A^, achieving a similar effect to that of BUB-1^L282A,V285A^. Previous data showed that substituting I for V in the LxxIxE motif decreases affinity and we believe in this case this renders the motif more dependent on phosphorylation. These results are included in the New Figure 6.

3) In addition to the previously reported mass spec data, we now show the localisation of the L(pS)PVSE motif by immunofluorescence with a newly generated phospho-specific antibody (New Figure 5G,H).

Reviewer #3:The study by Bel Borja and colleagues studies to which extent the PP2A phosphatase contributes to chromosome segregation in *C. elegans* anaphase. It finds that PP2A plays a crucial role in chromosome alignment and segregation. This role depends on the B56 regulatory subunits. Moreover, the author show that the recruitment of the B56 subunits to the spindle apparatus and kinetochores in anaphase depends on the Bub-1 kinase.The study explores a novel role of PP2A in *C. elegans* anaphase, and is generally of good technical quality (see below), yet at the same time feels a bit thin in terms of results. In particular several major claims of the discussion are not supported by the data. Moreover, it is not clear to which extent the reported experiments are reproducible.1) The authors do not indicate the number of independent experiments or the number of embryos on which the conclusions are based on. This information is essential to evaluate the reproducibility of the results. The authors use good statistical tests, but the reader must know whether the experiments are based on 5, 10 or 20 embryos.

We have now included all the information in the graphs. For intensity measurements, “N” is the number of experiments, and “n” is the number of oocytes. For the congression and alignment analysis, “N” is the number of spindles (=number of oocytes) and “n” is the number of bivalents analysed. In the latter analysis, information on the number of experiments used to obtain the data is stated in the figure legend.

2) The authors claim the Bub-1 acts on *C. elegans* anaphase by RECRUITING PP2A:B56 to the spindle apparatus. The presented experiments only show that the localization of the B56 depends on Bub-1, which is not the same thing. The authors could test by IP and in vitro pull-down that the claimed interaction is direct (in vitro), or that the proteins are part of the same complex (in vivo). This should be feasible since the authors already have performed IP experiments.

We thank the reviewer for raising this concern and we have now changed the wording when referring to this. We have performed in vitro binding experiments and present evidence for the direct interaction between the LxxIxE motif of BUB-1 and PPTR-2 (New Figures 4D and 5F). We expressed recombinant, full-length PPTR-2 in bacteria and performed fluorescence polarisation experiments using fluorescently labelled LxxIxE motif peptides. PPTR-2 bound the wild type sequence and that binding was abolished by the L282A,V285A mutations and enhanced when the peptide was phosphorylated at Serine 283.

3) The authors claim that PP2A:B56 counteracts Aurora-B during anaphase, but have no evidence for this claim. Does a Aurora-B hypomorph mutant attenuate the B56 loss phenotype? While a negative result would not mean that the model is wrong, a positive result would certainly help to bolster that claim.

As stated above, we think we are providing a significant advance on the mechanisms in place to regulate meiosis I by specific recruitment of PP2A/B56. While we are actively working on identifying PP2A substrates and its potential role in antagonising Aurora B, we believe this will require an in-depth analysis that will be the focus of another paper. We have now focused our discussion on our actual data and left this as an interesting, yet unsupported, hypothesis.